# TMPRSS2-mediated coronavirus spike activation and inhibition

Matthew McCallum [1,15,18], James Brett Case [2,3,18], Jack T. Brown[1,18], Young-Jun Park [1,4,18], Jimin Lee[1], Emmajay Sutherland [5], Anupriya Aggarwal[6], Cecily Gibson[1,4], Florian A. Lempp [7], Cameron Stewart [1], M. Alejandra Tortorici [1], Shilpa Sanapala[2], Jun Siong Low[8,9,16], Daniel Asarnow [1], Dana Bohan [7], Exequiel Dellota Jr.[7], Benjamin Merz[1], Bhavna Chawla[10], Swagata Kar[10], Antonio Lanzavecchia [11,12], Federica Sallusto[8,9], Nicholas M. Riley [5], Stuart Turville[6], Lisa Purcell[7,17], Michael S. Diamond [2,13,14] ✉ & David Veesler [1,4] ✉

The protease TMPRSS2 facilitates coronavirus infections, yet its mechanism of viral glycoprotein recognition remains unclear. Here we show that, following ACE2 engagement of the SARS-CoV-2 spike (S) inducing the early fusion intermediate conformation (E-FIC), TMPRSS2 cleaves the R815 $S_2'$ site and promotes fusogenic conformational changes leading to viral entry. We unveil TMPRSS2 recognition of $S_2'$, identify key residues modulating binding specificity and demonstrate that $S_2'$ site-directed broadly neutralizing antibodies target E-FIC and inhibit viral entry by blocking TMPRSS2 access. We computationally designed stabilized E-FIC as a vaccine candidate, overcoming the transient nature of this state. We describe a TMPRSS2-directed monoclonal antibody inhibiting several coronaviruses, including SARS-CoV-2 variants and protecting mice against SARS-CoV-2 challenge. These results outline the mechanistic role of TMPRSS2 and $S_2'$ site-directed antibodies in coronavirus entry.

The coronavirus spike (S) glycoprotein mediates infection by fusing viral and host membranes through large-scale conformational changes triggered by receptor binding and proteolysis. S is a metastable prefusion trimer anchored in the viral membrane[1–5], with its receptor-binding domains (RBDs) dynamically sampling conformations that balance host receptor engagement and immune evasion[6–9] (Fig. 1a). Early work showed that receptor binding modulates the sensitivity of S to proteases[10]. As proteases inactivate free pseudovirions but promote entry of cell-surface-bound ones[11], productive S cleavage appears coupled to receptor engagement. Moreover, receptor recognition leads to membrane insertion of the S fusion peptide and unmasking of the $S_2'$ site, allowing proteolytic cleavage to promote membrane fusion and viral entry[12–14]. Several ensuing studies suggested that receptor binding induces allosteric conformational changes enabling $S_2'$ cleavage[8,14–18] and binding of broadly neutralizing antibodies

targeting this site[19–21]. The recent structural elucidation of the S early fusion intermediate conformation (E-FIC) revealed that the $S_2'$ site becomes accessible in this state[22] (Fig. 1a). Subsequent S refolding to the postfusion state is believed to merge the two membranes to form a fusion pore (Fig. 1a), enabling genome delivery[23–29]. It remains unknown whether E-FIC is a structural checkpoint coupling receptor engagement to proteolytic activation.

Coronavirus S glycoproteins are processed by host proteases at up to two sites: the $S_1/S_2$ site (SARS-CoV-2 $R_{682}RAR_{685} \downarrow S_{686}$), present in a subset of coronaviruses and typically cleaved during S biogenesis, and the conserved $S_2'$ site (SARS-CoV-2 R815 $\downarrow$ S816), processed upon host cell entry[8,12,13,15,17,30]. The critical role of $S_2'$ cleavage is established by the deleterious effect of substitutions of the conserved $S_2'$ arginine on viral entry and fusion[13–15,31–34], the inhibitory activity of $S_2'$-directed antibodies[19–21] and cryo-electron tomography (cryo-ET) data showing that SARS-CoV-2

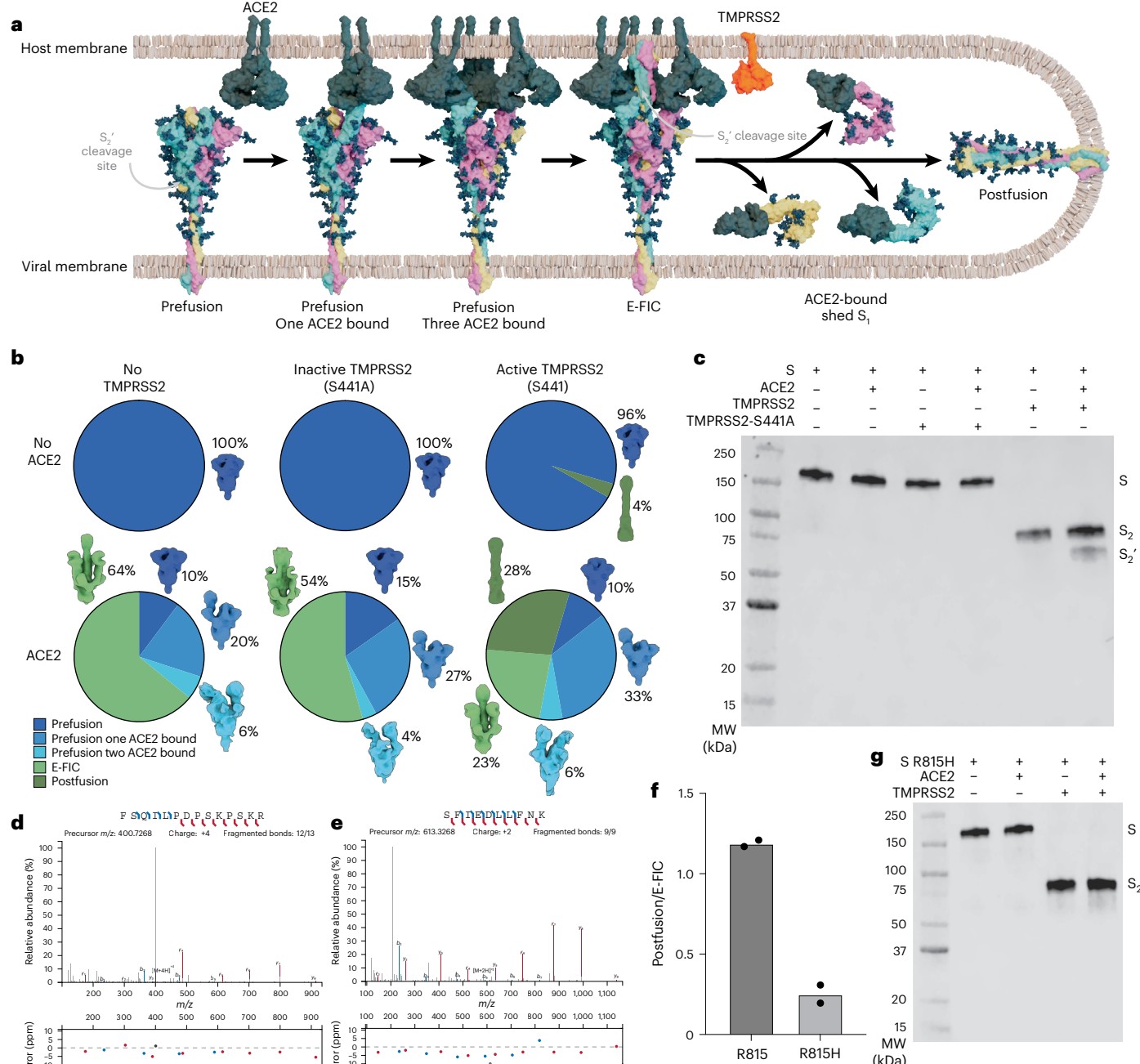

**Fig. 1 | TMPRSS2 cleaves SARS-CoV-2 S at position R815 to promote efficient E-FIC to postfusion transition. a**, Snapshots of key S conformations during viral entry: prefusion S (based on PDB 6VXX), prefusion S bound to one ACE2 receptor (based on PDB 7A94), prefusion S bound to three ACE2 receptors (based on PDB 7A97), E-FIC (PDB 8Z7P), postfusion S (PDB 8FDW) and TMPRSS2 (based on PDB 7MEQ). The prefusion S conformation bound to two ACE2 receptors is not shown for clarity. **b**, Single-particle EM analysis of conformations detected for negatively stained SARS-CoV-2 $S_{ecto}$ at a concentration of 0.65 μM incubated (or not) with 2 μM of the monomeric human ACE2 (peptidase) ectodomain for 5 min before incubation (or not) with the TMPRSS2 ectodomain at a concentration of 0.2 μM for 45 min. All steps were carried out at 4 °C. Pie charts show the distribution of selected particles in each conformation and corresponding 3D reconstructions for the indicated datasets. **c**, Western blot of SARS-CoV-2 $S_{ecto}$ incubated with ACE2 and/or TMPRSS2 (as described in **b**) detected with the stem helix-directed B6

(ref. 66) primary antibody. The $S_2'$ cleavage product for this construct is expected at 64 kDa (49 kDa based on the protein sequence and six 2.5-kDa N-linked glycans). **d,e**, The b/y ion mass spectra for the $F_{802}$SQILPDPSKPSKR$_{815}$ (**d**) and $S_{816}$FIEDLLFNK$_{825}$ (**e**) peptides (corresponding to TMPRSS2-mediated cleavage at position R815) detected upon SARS-CoV-2 $S_{ecto}$ incubation with ACE2 and active (S441) TMPRSS2 as described in **b**. **f**, Ratio of particle images assigned to postfusion S and E-FIC upon classification of single-particle EM data of negatively stained SARS-CoV-2 $S_{ecto}$ and SARS-CoV-2 $S_{ecto}$-R815H incubated with ACE2 and active (S441) TMPRSS2, as described in **b**. Each data point corresponds to one biological replicate. **g**, Western blot analysis of SARS-CoV-2 $S_{ecto}$-R815H incubated with ACE2 and/or TMPRSS2 (as described in **b**) detected with the stem helix-directed B6 (ref. 66) primary antibody. Western blot analyses in **c**,**g** were carried out with n = 2 biological replicates using independently produced batches of proteins (Supplementary Figs. 1, 2, 15 and 16).

fusion depends on $S_2'$ proteolysis[26]. Consequently, host receptors and proteases are determinants of coronavirus tropism and pathogenesis.

The transmembrane serine protease TMPRSS2 has a central role in the entry of coronaviruses[32,34–49] and influenza viruses[50,51]. Although

multiple proteases can mediate $S_2'$ cleavage, including endosomal proteases[14,32,36,52,53], the plasma membrane localization of TMPRSS2 makes it particularly effective for cell entry and bypass endosomal restriction factors[38,54–57]. The importance of TMPRSS2 for coronavirus

entry is further underscored by its direct use as an entry receptor by the endemic human coronavirus HKU1 (refs. 58–61). As a key conserved host factor, TMPRSS2 represents an attractive therapeutic target. However, existing TMPRSS2 inhibitors often lack specificity, possibly limiting their clinical use because of off-target effects[42,48,62–64]. Selective blockade of TMPRSS2-dependent virus activation could transform how we combat viral threats.

Despite the importance of $S_2'$ site cleavage for coronavirus infection, the molecular basis of protease recognition, the specific S conformation with which host proteases interact and the conformational consequences of TMPRSS2-mediated S cleavage are unknown. This knowledge gap limits our understanding of membrane fusion and hinders efforts to target this step. Here, we show that TMPRSS2 cleavage of E-FIC promotes fusion triggering, we computationally design a stabilized E-FIC vaccine candidate and we establish a mechanistic framework for targeting TMPRSS2.

## Results

### TMPRSS2 induces the conformational transition from E-FIC to postfusion S

To elucidate the specific contribution of TMPRSS2-mediated cleavage to membrane fusion, we evaluated its impact on SARS-CoV-2 S in various conditions recapitulating the viral entry process. A wild-type-like prefusion SARS-CoV-2 Wu/G614 S ectodomain trimer harboring a $S_{682}GAR_{685}$ $S_1/S_2$ sequence (designated SARS-CoV-2 $S_{ecto}$) was incubated with the monomeric human ACE2 peptidase domain and the human TMPRSS2 ectodomain to analyze the distribution of S conformations by single-particle electron microscopy (EM) analysis of negatively stained samples. In the absence of ACE2, SARS-CoV-2 $S_{ecto}$ adopts the prefusion conformation with or without added catalytically inactive (S441A) or active (S441) TMPRSS2 (Fig. 1b,c and Supplementary Figs. 1–4). Although S metastability was enhanced upon cleavage at the $S_1/S_2$ junction in presence of S441 TMPRSS2 (ref. 65), 96% of particles remained in the prefusion state, consistent with the wild-type SARS-CoV-1 S ectodomain trimer predominantly remaining in the prefusion state in the absence of ACE2, with or without trypsin[8]. These data underscore that the $S_2'$ cleavage site remains inaccessible in the prefusion conformation.

Incubation of SARS-CoV-2 $S_{ecto}$ with ACE2 promoted large-scale conformational changes and transition of most particles to E-FIC arranged as trefoil-like rosettes through clustering of the hydrophobic fusion peptides (Fig. 1b,c and Supplementary Figs. 1, 2, 5 and 6). In these conditions, apo prefusion S, prefusion S bound to one ACE2 receptor, prefusion S bound to two ACE2 receptors and E-FIC accounted for approximately 10%, 20%, 6% and 64% of selected particle images. Addition of catalytically inactive (S441A mutant) TMPRSS2 to ACE2-bound S did not substantially affect the relative distribution of S conformations detected, compared to incubation with ACE2 alone, with approximately 15%, 27%, 4% and 54% of particles corresponding to apo prefusion S, prefusion S bound to one ACE2 receptor, prefusion S bound to two ACE2 receptors and E-FIC, respectively (Fig. 1b,c and Supplementary Figs. 1, 2, 7 and 8).

However, addition of catalytically active (S441) TMPRSS2 induced the conformational transition from E-FIC to postfusion S, visualized as rosettes of rod-shaped trimers clustered by the fusion peptides[23–25] (Fig. 1b,c and Supplementary Figs. 1, 2, 9 and 10). This transition was accompanied by a second S cleavage event corresponding to the SARS-CoV-2 $S_2'$ site ($R_{815} \downarrow S_{816}$), as supported by the detection of a ~65-kDa band by western blotting with the stem helix-directed monoclonal antibody (mAb) B6 (ref. 66) but not the $S_2'$-directed mAb 76E1 (ref. 20), with the latter mAb requiring the $S_2'$ R815 residue for binding (Fig. 1c and Supplementary Figs. 1 and 2). Mass spectrometry (MS) analysis confirmed that TMPRSS2 cleaves E-FIC at position R815 with detection of the $F_{802}SQILPDPSKPSKR_{815}\downarrow$ peptide solely in the presence of ACE2 and catalytically active (S441) TMPRSS2 and the $S_{816}FIEDLL-FNK_{825}$ peptide in the presence of S441 TMPRSS2 with or without ACE2.

The MS/MS data show markedly more effective cleavage of E-FIC (in the presence of ACE2) than prefusion S (in the absence of ACE2), consistent with exposure of the $S_2'$ cleavage site in E-FIC but not prefusion S (Fig. 1d,e, Extended Data Fig. 1a,b and Supplementary Figs. 11 and 12). In the presence of ACE2 and S441 TMPRSS2, apo prefusion S, prefusion S bound to one ACE2 receptor, prefusion S bound to two ACE2 receptors, E-FIC and postfusion S accounted for approximately 10%, 33%, 6%, 23% and 28% of selected particle images, respectively. We also detected ACE2-bound free $S_1$ protomers[67], shed during the refolding of E-FIC to postfusion S (Supplementary Figs. 13 and 14). The detection of a large fraction of postfusion S by EM solely in the presence of ACE2 and active TMPRSS2 concurs with the prominent ~55-kDa proteinase-K-resistant band (postfusion six-helix bundle[8,12]) observed in the same conditions by western blotting (Supplementary Figs. 1 and 2). The E-FIC fraction decreased more than twofold in the presence of ACE2 and active TMPRSS2 relative to S incubated with only ACE2 or ACE2 and inactive TMPRSS2, whereas the prefusion S fractions remained roughly constant. The reduced E-FIC abundance concurs with TMPRSS2 specifically cleaving this conformational state, promoting the conformational changes leading to postfusion S.

To further confirm that the conformational transition from E-FIC to postfusion S results from TMPRSS2-mediated cleavage at position R815, we analyzed the distribution of conformations of SARS-CoV-2 $S_{ecto}$ harboring the R815H substitution in the presence of ACE2 and TMPRSS2. We found that R815H S folded as a prefusion trimer in the absence of ACE2 and as E-FIC in the presence of ACE2 (Fig. 1f and Supplementary Figs. 15–20). In contrast to R815 S, we did not detect $S_2'$ cleavage of R815H S, as TMPRSS2 preferentially cleaves after arginine (or lysine) residues. Consequently, R815H S had a markedly reduced ratio of postfusion S to E-FIC in the presence of ACE2 and TMPRSS2, relative to R815 S (Fig. 1e,f and Supplementary Figs. 15–20). The detection of postfusion S without $S_2'$ cleavage is consistent with the ability of large excess of ACE2 or functional antibody mimics of ACE2 to induce refolding of prefusion SARS-CoV-1 S and SARS-CoV-2 S without exogenous protease addition[8,23,24,68–73]. However, the low amounts of ACE2 present on relevant target cells[74] and the enhancement of fusion, entry and replication observed in the presence of membrane-associated or extracellular proteases underscore the key role of TMPRSS2 and of the conserved $S_2'$ arginine residue (Extended Data Fig. 1c) for efficient E-FIC refolding to postfusion S[11,34–36,41,47,75–78]. Collectively, these results show that TMPRSS2 specifically recognizes and cleaves E-FIC at position R815, promoting efficient fusion of the host membrane (in which the fusion peptide is embedded) and the viral membrane (tethered to the S transmembrane domain) leading to viral entry.

### Molecular basis of TMPRSS2 recognition of the conserved coronavirus $S_2'$ site

To understand S glycoprotein engagement by TMPRSS2, we evaluated binding to a panel of coronaviruses $S_2'$ peptides spanning all four genera using biolayer interferometry (BLI). The TMPRSS2-S441A ectodomain bound more effectively to the HCoV-NL63-$S_2'$ peptide than to the HCoV-229E and PDCoV $S_2'$ peptides (Fig. 2a and Supplementary Table 1). Weak to no binding to the other peptides was observed when using 1 μM TMPRSS2-S441A, whereas detectable binding to all peptides, except the HKU1 $S_2'$ peptide, occurred with 5 μM TMPRSS2-S441A (Fig. 2a and Extended Data Fig. 1d). Further characterization of TMPRSS2-S441A binding to the HCoV-NL63-$S_2'$ peptide revealed fast association and dissociation kinetics, yielding an equilibrium dissociation constant ($K_D$) of approximately 500 nM (Fig. 2b and Supplementary Table 2). These findings indicate that TMPRSS2 recognizes the $S_2'$ region with modest affinity, consistent with the lack of TMPRSS2-bound SARS-CoV-2 $S_{ecto}$ trimers detected in our EM analysis (Fig. 1) and with the transient nature of enzyme–substrate interactions.

To visualize TMPRSS2-mediated recognition of a coronavirus $S_2'$ site and overcome the transient peptide binding, we leveraged

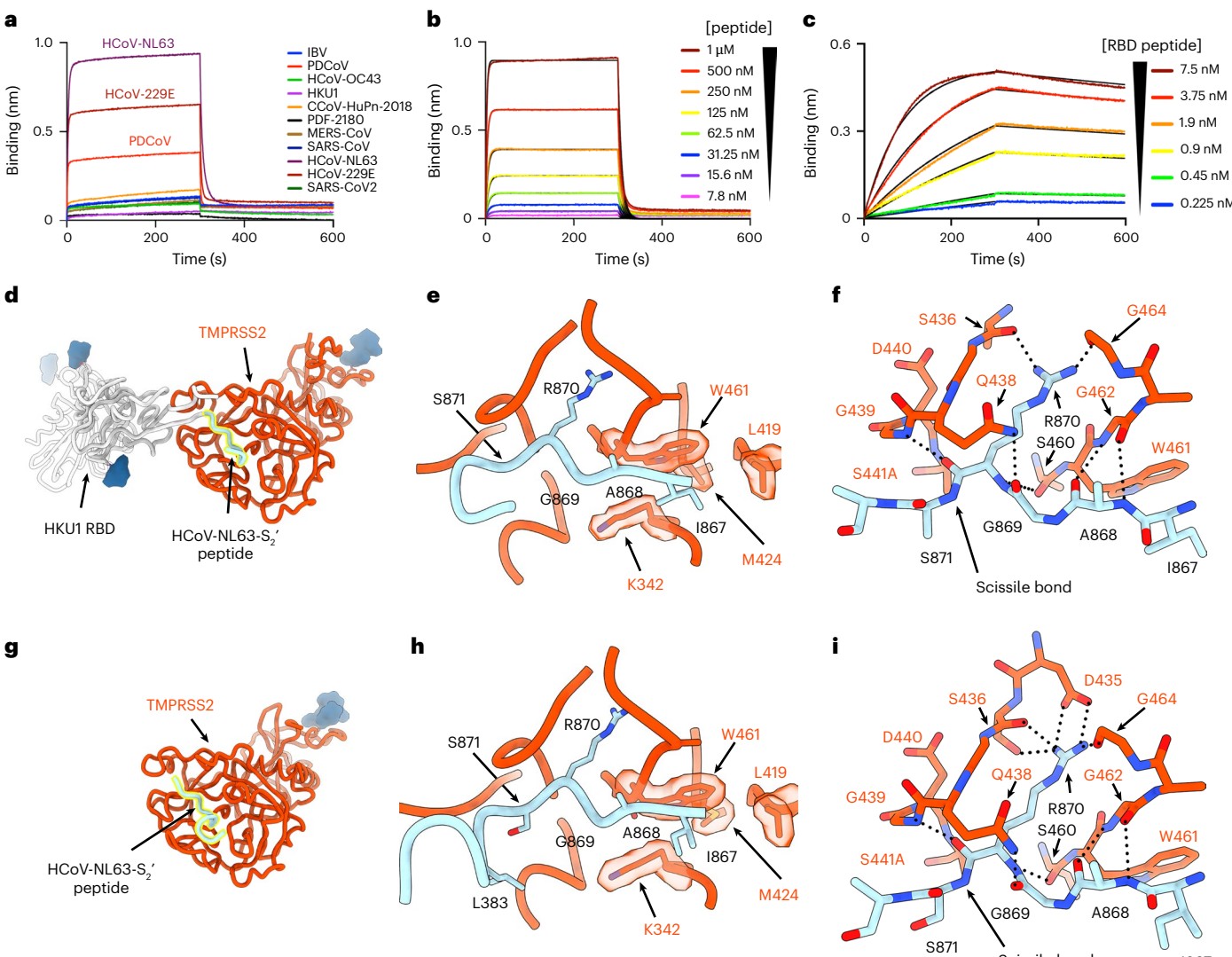

**Fig. 2 | Molecular basis of TMPRSS2 recognition of the conserved coronavirus S₂′ site. a**, Binding of the human TMPRSS2-S441A (catalytically inactive) ectodomain at a concentration of 1 µM to a panel of biotinylated coronavirus S₂′ peptides immobilized at the surface of BLI SA biosensors. **b,c**, Evaluation of binding kinetics and affinity of the human TMPRSS2-S441A ectodomain at various concentrations to the biotinylated HCoV-NL63-S₂′ peptide (**b**) or to the biotinylated HCoV-NL63-S₂′/HKU1-RBD (**c**) immobilized at the surface of BLI SA biosensors. Global fit using a 1:1 binding model is shown as black lines in **b,c**. BLI data shown in **a–c** correspond to one representative of two biological replicates, each using independently produced batches of proteins. **d,g**, Ribbon diagrams of the HCoV-NL63-S₂′ (cyan with yellow silhouette)/HKU1-RBD (semitransparent gray) bound to TMPRSS2-S441A (orange) structure (**d**) and of the HCoV-NL63-S₂′ peptide-bound (cyan with yellow silhouette) TMPRSS2-S441A (orange) structure (**g**). N-linked glycans are rendered as blue surfaces. **e,f,h,i**, Close-up views of the HCoV-NL63-S₂′ peptide bound to the TMPRSS2 active site for the structures shown in **d,g**. The HKU1 RBD fusion (**e,f**) and the H1H7 Fab and anti-kappa light chain nanobody used for structural determination (**h,i**) are omitted for clarity. Select residues and polar interactions are emphasized as semitransparent surfaces and shown as dashed black lines, respectively.

our structural understanding of HKU1 use of TMPRSS2 as a receptor[60] to engineer a fusion of the HCoV-NL63-S₂′ sequence to the HKU1 RBD, which we designate HCoV-NL63-S₂′/HKU1-RBD. TMPRSS2-S441A interacted strongly with HCoV-NL63-S₂′/HKU1-RBD, with a $K_D$ of approximately 0.2 nM (Fig. 2c), corresponding to a three orders of magnitude enhancement compared to the free HCoV-NL63-S₂′ peptide (Fig. 2c and Supplementary Table 2). Using single-particle cryo-electron microscopy (cryo-EM), we determined a structure of the TMPRSS2-S441A-bound HCoV-NL63-S₂′/HKU1-RBD at a resolution of 3.3 Å, illuminating coronavirus S₂′ substrate recognition (Fig. 2d–f, Extended Data Fig. 2a and Table 1). To confirm these findings, we determined a single-particle cryo-EM structure of TMPRSS2-S441A bound to the free HCoV-NL63-S₂′ peptide (using a saturating peptide concentration and the H1H7 Fab with anti-kappa light chain nanobody, discussed subsequently) at 2.8-Å resolution, revealing a close agreement with the

HCoV-NL63-S₂′/HKU1-RBD structure (Fig. 2g–i, Extended Data Fig. 2b and Table 1). The maps resolve the HCoV-NL63 I₈₆₇AGR ↓ SALE₈₇₄ S₂′ peptide burying ~640 Å² at the interface with the TMPRSS2 active site through a combination of polar interactions and shape complementarity. HCoV-NL63 residues at positions P1 and P4 account for the majority of the S₂′ peptide buried surface area upon binding. Specifically, the S₂′ Arg₈₇₀ (P1; equivalent to SARS-CoV-2 R815) forms a salt bridge with Asp₄₃₅ and is hydrogen-bonded to TMPRSS2 S436, G439, D440, S441, S460 and G464, positioning the scissile bond (↓) in close proximity to the TMPRSS2 S441 catalytic residue (substituted to A441; Fig. 2e,f). The S₂′ I867 (P4) side chain inserts into a hydrophobic cavity defined by TMPRSS2 K342, L419, M424 and W461, promoting extensive van der Waals interactions (Fig. 2e,f).

Integrating protein binding and structural data suggest that residues at positions P2 and P4 of the S₂′ peptide are key modulators

**Table 1 | Cryo-EM data collection, refinement and validation statistics**

| | HCoV-NL63-S$_2$'/ HKU1-RBD+TMPRSS2-S441A (EMD-70721), (PDB 9OPQ) | HCoV-NL63-S$_2$'+TMPRSS2-S441A+H1H7 Fab + anti-kappa-nanobody (EMD-73786), (PDB 9Z3J) | H1H7 Fab+anti-kappa-nanobody+ TMPRSS2-S441A (EMD-70722), (PDB 9OPR) |
|---|---|---|---|
| **Data collection and processing** | | | |
| Magnification | ×130,000 | ×130,000 | ×130,000 |
| Voltage (kV) | 300 | 300 | 300 |
| Electron exposure (e⁻ per Å$^2$) | 63 | 60 | 63 |
| Defocus range (µm) | 0.3–2.0 | 0.3–2.0 | 0.3–2.0 |
| Pixel size (Å) | 0.829 | 0.829 | 0.843 |
| Symmetry imposed | $C_1$ | $C_1$ | $C_1$ |
| Initial particle images (no.) | 1,299,323 | 2,000,257 | 314,213 |
| Final particle images (no.) | 163,873 | 270,089 | 117,441 |
| Map resolution (Å) | 3.3 | 2.8 | 3.2 |
| FSC threshold | 0.143 | 0.143 | 0.143 |
| **Refinement** | | | |
| Initial model used (PDB code) | 8VGT | 9OPR | 7MEQ |
| Model resolution (Å) | 3.4 | 2.9 | 3.3 |
| FSC threshold | 0.5 | 0.5 | 0.5 |
| Map sharpening $B$ factor (Å$^2$) | −110 | −100 | −106 |
| Model composition | | | |
| Nonhydrogen atoms | 4,572 | 6,814 | 6,517 |
| Protein residues | 634 | 895 | 886 |
| Ligands | 6 | 3 | 2 |
| $B$ factors (Å$^2$) | | | |
| Protein | 33.08 | 13.40 | 53.14 |
| Ligand | 78.92 | 21.45 | 110.70 |
| Root-mean-square deviations | | | |
| Bond lengths (Å) | 0.004 | 0.010 | 0.004 |
| Bond angles (°) | 0.536 | 0.975 | 0.551 |
| **Validation** | | | |
| MolProbity score | 1.24 | 0.96 | 0.95 |
| Clashscore | 4.65 | 1.93 | 1.87 |
| Poor rotamers (%) | 0.28 | 0.31 | 0.00 |
| Ramachandran plot | | | |
| Favored (%) | 98.22 | 98.64 | 98.51 |
| Allowed (%) | 1.78 | 1.36 | 1.49 |
| Disallowed (%) | 0 | 0 | 0 |

of binding specificity and affinity along with the conserved P1 arginine. The extensive contacts between HCoV-NL63 P4 residue I867 and TMPRSS2 would be preserved for HCoV-229E and PDCoV, which respectively harbor comparable valine and leucine P4 residues (Extended Data Fig. 1c). Moreover, conservation of HCoV-NL63 P2 residue G869 with HCoV-229E and PDCoV would promote optimal fitting with TMPRSS2 for all three viruses (Extended Data Fig. 1c). A small side chain, such as HCoV-NL63 A872, which is also conserved for HCoV-229 and PDCoV, likely favors binding at position P2'. Accordingly, TMPRSS2 recognition of the HCoV-NL63, HCoV-229 and PDCoV S$_2$' peptides was characterized by greater binding affinity relative to the other orthologs tested (Fig. 2a, Supplementary Table 1 and Extended Data Fig. 1d). The recently described A07 nanobody, which inserts its complementarity-determining region 3 (CDR3) loop directly into the TMPRSS2 catalytic site[59], harbors similar residues at positions equivalent to the P4 I867 (A07 V102), P2 G869 (A07 G104) and R870 P1

(A07 R105) residues, pointing to convergent selection of these residues for TMPRSS2 binding (Extended Data Fig. 1e,f). Alignment of the TMPRSS2-bound HCoV-NL63 residues 867–870 with the corresponding residues in SARS-CoV-2 S E-FIC[22] (residues 812–815, which are poorly resolved in E-FIC structures) illustrates how TMPRSS2 could access its target site with minor structural rearrangements because of the nearby ACE2-bound RBDs. Our data provide a blueprint of the interactions that lead to TMPRSS2 recognition and activation of coronavirus S glycoproteins.

### E-FIC is the target of S$_2$' site-directed antibodies

The identification of E-FIC as the target of TMPRSS2 led us to hypothesize that S$_2$' site-directed antibodies also recognize this conformational state. To investigate this possibility, we structurally characterized a complex formed between SARS-CoV-2 E-FIC and the VN01H1 mAb, which neutralizes several human alphacoronaviruses

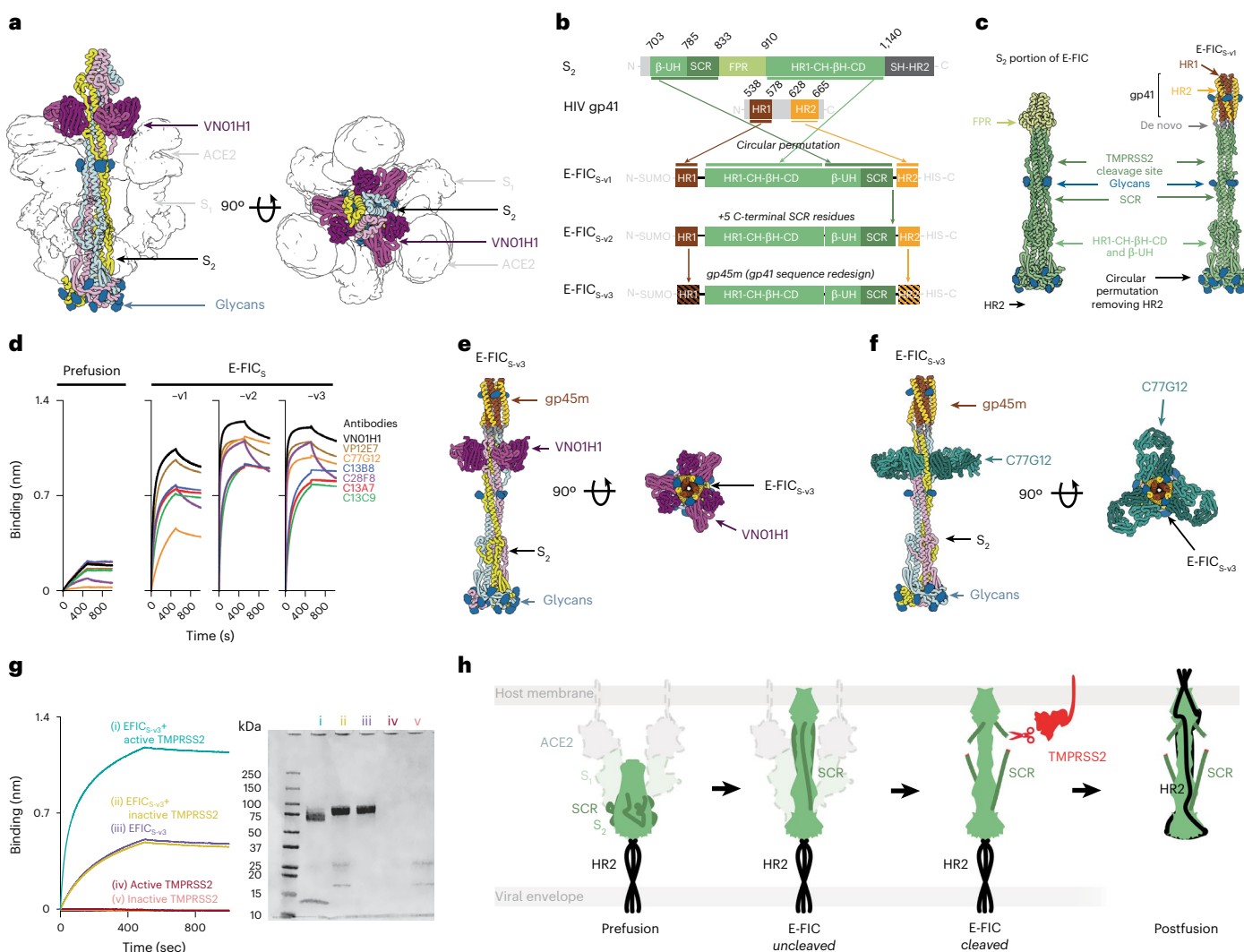

**Fig. 3 | SARS-CoV-2 E-FIC is the target of S₂′ site-directed antibodies and a vaccine candidate. a**, Ribbon diagrams in two orthogonal orientations of the VN01H1 Fab-bound SARS-CoV-2 E-FIC structure. Rendering used a composite model obtained from local cryo-EM refinements of the viral membrane-proximal and membrane-distal regions. Each S protomer is colored distinctly, *N*-linked glycans are shown as blue surfaces and an outline of the S₁ subunit and ACE2 from the cryo-EM map are shown. VN01H1 Fab heavy and light chains are colored purple and magenta, respectively. **b**, Design strategy for stabilizing SARS-CoV-2 E-FIC (E-FICₛ). SARS-CoV-2 β-UH, β718–728 and upstream helix (residues 703–785); SCR, S₂′ site cleavage region (residues 786–833); FPR, fusion peptide region (residues 834–910 comprising the preceding region and actual fusion peptide); HR1-CH-βH-CD, heptad repeat motif 1, central helix, β-hairpin and connector domain (residues 911–1140); SH-HR2, stem helix and heptad repeat motif 2.

**c**, Comparison of the E-FIC (PDB 8Z7G) and E-FICₛ structures. **d**, Binding of prefusion SARS-CoV-2 HexaPro S (0.05 mg ml⁻¹) and E-FICₛ-v1, E-FICₛ-v2 and E-FICₛ-v3 (0.1 mg ml⁻¹) to a panel of S₂′ site-directed antibodies immobilized on anti-human Fc biosensors analyzed by BLI. **e,f**, Cryo-EM structures of VN01H-bound (**e**) or C77G12-bound (**f**) E-FICₛ-v3 colored as in **a**. **g**, Left, binding of E-FICₛ-v3 preincubated with active (S441) TMPRSS2 (cyan) or inactive (A441) TMPRSS2 (gold) or not (purple) to biotinylated SARS-CoV-2 HR2 peptide immobilized on SA biosensors analyzed by BLI. Controls with lack of binding of active TMPRSS2 (maroon) or inactive TMPRSS2 (pink) to HR2, as well as all corresponding conditions without immobilized HR2 (orange), are shown. Right, SDS–PAGE of the corresponding samples (*n* = 1 biological replicate shown). **h**, Schematic model illustrating the S conformational transition. S₂′ site (R815) cleavage enables HR2 (dark green) to efficiently reach its binding site on HR1 leading to transition to postfusion S.

and betacoronaviruses[19]. We used single-particle cryo-EM and local refinement workflows to determine a structure of the complex at 3.3-Å resolution, revealing the presence of three VN01H1 Fab fragments bound to the E-FIC trimer (Fig. 3a, Extended Data Fig. 3a and Table 2). VN01H1 uses both heavy and light chains to recognize the S₂′ region, which adopts an extended helical conformation relative to that of prefusion S[4]. The contacts formed between VN01H1 and E-FIC are similar to the crystal structure of the Fab fragment bound to the S₂′ peptide[19], involving extensive interactions with R815, E819 and F823 (Fig. 3a and Extended Data Fig. 4a). However, in the VN01H1-bound E-FIC structure, the Fab also contacts residues 825–830 and the HR1 region of a neighboring protomer (Extended Data Fig. 4a). These findings show how VN01H1 and similar S₂′ site-directed antibodies would sterically

hinder TMPRSS2 binding to E-FIC (through recognition of an overlapping site on S) and, in turn, proteolytic activation of membrane fusion and viral entry[19–21].

## Computational design of a stable E-FIC

Given that the S₂′ site is exposed in E-FIC and the target of both TMPRSS2 and the most broadly neutralizing coronavirus antibodies, we aimed to develop a strategy to generate a stable E-FIC construct that would overcome its transient and aggregation-prone nature. Circular permutation of the N-terminal and C-terminal extremities of the SARS-CoV-2 S₂ subunit and fusion to HIV gp41 HR1/HR2 and to the designed oP3h trimer[79] yielded a construct designated E-FICₛ-v1 that was predicted to fold as E-FIC in the absence of the S₁ subunit and receptor (Fig. 3b,c). To assess

**Table 2 | Cryo-EM data collection, refinement and validation statistics**

| | E-FIC+VNO1H1 Fab (global refinement) (EMD-75233) | E-FIC+VNO1H1 Fab (Fab local refinement) (EMD-73656) (PDB 9YYU) | E-FIC+VNO1H1 Fab (S$_2$ local refinement) (EMD-73657) (PDB 9YYV) | E-FIC$_{s-v1}$ (EMD-73787) (PDB 9Z3K) | E-FIC$_{s-v3}$+VNO1H1 Fab (global refinement) (EMD-75721) | E-FIC$_{s-v3}$+VNO1H1 Fab (Fab local refinement) (EMD-75694) (PDB 11HK) | E-FIC$_{s-v3}$+VNO1H1 Fab (S$_2$ local refinement) (EMD-75695) (PDB 11HL) | E-FIC$_{s-v3}$+C77G12 Fab (global refinement) (EMD-75722) | E-FIC$_{s-v3}$+C77G12 Fab (Fab local refinement) (EMD-75705) (PDB 11HW) | E-FIC$_{s-v3}$+C77G12 Fab (S$_2$ local refinement) (EMD-75697) (PDB 11HN) |
|---|---|---|---|---|---|---|---|---|---|---|
| **Data collection and processing** | | | | | | | | | | |
| Magnification | ×130,000 | ×130,000 | ×130,000 | ×130,000 | ×130,000 | ×130,000 | ×130,000 | ×130,000 | ×130,000 | ×130,000 |
| Voltage (kV) | 300 | 300 | 300 | 300 | 300 | 300 | 300 | 300 | 300 | 300 |
| Electron exposure (e$^-$ per Å$^2$) | 60 | 60 | 60 | 60 | 60 | 60 | 60 | 60 | 60 | 60 |
| Defocus range (μm) | 0.2–3.0 | 0.2–3.0 | 0.2–3.0 | 0.3–2.0 | 0.3–2.0 | 0.3–2.0 | 0.3–2.0 | 0.3–2.0 | 0.3–2.0 | 0.3–2.0 |
| Pixel size (Å) | 0.835 | 0.835 | 0.835 | 0.829 | 0.829 | 0.829 | 0.829 | 0.829 | 0.829 | 0.829 |
| Symmetry imposed | C$_3$ | C$_3$ | C$_3$ | C$_3$ | C$_3$ | C$_3$ | C$_3$ | C$_3$ | C$_3$ | C$_3$ |
| Initial particle images (no.) | 1,511,008 | 1,511,008 | 1,511,008 | 3,105,263 | 291,929 | 121,952 | 121,952 | 434,056 | 434,056 | 434,056 |
| Final particle images (no.) | 389,200 | 389,200 | 389,200 | 114,104 | 121,952 | 121,952 | 121,952 | 225,859 | 225,859 | 225,859 |
| Map resolution (Å) | 3.8 | 3.3 | 3.3 | 3.3 | 2.9 | 2.7 | 2.8 | 2.9 | 2.8 | 2.7 |
| FSC threshold | 0.143 | 0.143 | 0.143 | 0.143 | 0.143 | 0.143 | 0.143 | 0.143 | 0.143 | 0.143 |
| **Refinement** | | | | | | | | | | |
| Initial model used (PDB code) | | 8Z7P, 7SKZ | 8Z7P | N/A | | 7SKZ | N/A | | 7UOA | N/A |
| Model resolution (Å) | | 3.4 | 3.4 | 3.5 | | 3.0 | 3.0 | | 2.8 | 3.0 |
| FSC threshold | | 0.5 | 0.5 | 0.5 | | 0.5 | 0.5 | | 0.5 | 0.5 |
| Map sharpening B factor (Å$^2$) | | −81 | −100 | −92 | | −84 | −87 | | −87 | −87 |
| **Model composition** | | | | | | | | | | |
| Nonhydrogen atoms | | 8,739 | 6,426 | 8,448 | | 11,595 | 5,208 | | 9,288 | 5,118 |
| Protein residues | | 1,215 | 852 | 1,212 | | 1,680 | 690 | | 1,254 | 675 |
| Ligands | | 3 | 15 | 18 | | 6 | 12 | | 6 | 12 |
| **B factors (Å$^2$)** | | | | | | | | | | |
| Protein | | 54.75 | 43.07 | 79.19 | | 30.96 | 21.16 | | 40.22 | 37.42 |
| Ligand | | 116.71 | 97.19 | 88.68 | | 37.44 | 34.79 | | 66.95 | 50.37 |
| **Root-mean-square deviations** | | | | | | | | | | |
| Bond lengths (Å) | | 0.002 | 0.002 | 0.012 | | 0.011 | 0.011 | | 0.013 | 0.012 |
| Bond angles (°) | | 0.413 | 0.479 | 0.947 | | 0.979 | 0.967 | | 1.016 | 1.051 |
| **Validation** | | | | | | | | | | |
| MolProbity score | | 1.39 | 1.18 | 0.64 | | 0.86 | 0.94 | | 0.97 | 1.05 |
| Clashscore | | 4.79 | 3.95 | 0.39 | | 1.32 | 1.51 | | 2.00 | 1.63 |
| Poor rotamers (%) | | 0.4 | 0 | 0.00 | | 0.00 | 0.00 | | 0.39 | 0.00 |
| **Ramachandran plot** | | | | | | | | | | |
| Favored (%) | | 97.24 | 98.56 | 98.48 | | 99.07 | 97.79 | | 98.78 | 97.26 |
| Allowed (%) | | 2.76 | 1.44 | 1.52 | | 0.93 | 2.21 | | 1.22 | 2.74 |
| Disallowed (%) | | 0 | 0 | 0 | | 0 | 0 | | 0 | 0 |

whether E-FIC$_{S-v1}$ folds as designed, we determined a single-particle cryo-EM structure at 3.3-Å resolution showing that it adopts an E-FIC structure with HR1 refolded to form a central three-helix bundle flanked by residues 770–828 (Fig. 3c, Extended Data Fig. 3b and Table 2). In this conformation, the S$_2'$ site is peripherally exposed and weakly resolved in the cryo-EM map, likely promoting accessibility to protease (similar to the S$_1$/S$_2$ cleavage site[4,8]) (Fig. 3c). E-FIC$_{S-v1}$ bound markedly more effectively to a panel of S$_2'$ site-directed antibodies than prefusion HexaPro S[80], confirming exposure of the S$_2'$ site—a hallmark of this conformation[22] (Fig. 3d). However, residues downstream of the S$_2'$ cleavage site were shifted relative to bona fide E-FIC. To promote native positioning, we included five additional residues from the S$_2'$ site cleavage region (up to residue 833), yielding E-FIC$_{S-v2}$ (Fig. 3b and Extended Data Fig. 4d), which showed improved binding to S$_2'$ site-directed antibodies, most notably C77G12 (Fig. 3d). To enable possible future use as a human vaccine, gp41 HR1/HR2 were replaced with the computationally redesigned equine infectious anemia virus gp41 homolog (designated gp45m), yielding E-FIC$_{S-v3}$, which bound S$_2'$ site-directed antibodies comparably to E-FIC$_{S-v2}$ (Fig. 3b,d and Extended Data Fig. 4d,e). Structural analysis of E-FIC$_{S-v3}$ in complex with the VN01H1 Fab recapitulated the interactions described above (Fig. 3e, Extended Data Figs. 3c and 4a,b and Table 2). Moreover, a cryo-EM structure of C77G12 Fab-bound E-FIC$_{S-v3}$ also revealed quaternary interactions that could not be visualized in the crystal structure obtained with the free S$_2'$ peptide[19], including cross-protomer contacts and a rare homotypic binding mode thought to enhance affinity and B cell activation[81] (Fig. 3f, Extended Data Figs. 3d and 4c and Table 2). These results demonstrate that E-FIC$_{S-v3}$ recapitulates native E-FIC antibody recognition and paves the way for streamlining future structural and immunological studies.

## S$_2'$ site cleavage promotes refolding to the postfusion state

The position of the S$_2'$ site in E-FIC provides a mechanistic rationale linking cleavage and S refolding; residues 770–830 occupy the HR2 binding site on HR1, sterically preventing formation of the HR1/HR2 six-helix bundle characteristic of postfusion S[8,23,25]. To functionally evaluate this model, we investigated binding of E-FIC$_{S-v3}$ to immobilized HR2 peptide by BLI. We observed that binding of E-FIC$_{S-v3}$ to HR2 was markedly enhanced by preincubation with active (S441) TMPRSS2, because of proteolytic processing, relative to untreated E-FIC$_{S-v3}$ or E-FIC$_{S-v3}$ preincubated with inactive (A441) TMPRSS2 (Fig. 3g). These findings indicate that cleavage of E-FIC at the S$_2'$ site promotes efficient access of HR2 to its binding site on HR1 and refolding to postfusion S (Fig. 3h). Accordingly, cryo-ET imaging of authentic SARS-CoV-2 virions mixed with ACE2-harboring pseudoviruses showed that S$_2'$ cleavage is required for membrane fusion to take place[26]. We propose that transition through E-FIC allows the fusogenic conformational changes of multiple S trimers on a virion to take place in a coordinated manner with TMPRSS2 cleavage promoting efficient refolding to postfusion S and in turn viral entry.

## The H1H7 mAb inhibits TMPRSS2-mediated viral infection

H1H7017N (abbreviated to H1H7) is an mAb that was isolated from mice immunized with an antigen comprising the TMPRSS2 low-density lipoprotein receptor class A and scavenger receptor cysteine-rich (SRCR) domains (without the protease domain)[82]. We recombinantly produced H1H7 in a human IgG1 backbone and evaluated binding to immobilized TMPRSS2-S441A using BLI. H1H7 IgG bound TMPRSS2 with a high apparent affinity (apparent $K_D \leq 1$ pM) and its Fab fragment bound TMPRSS2 with a 1:1 affinity ($K_D$) of 0.8 nM, both characterized by slow off rates (Extended Data Fig. 5a,b and Supplementary Table 3). To determine the effect of H1H7 on viral infection, we first performed studies with a vesicular stomatitis virus (VSV) pseudotyped with the SARS-CoV-2 Wuhan-Hu-1 S glycoprotein. H1H7 inhibited SARS-CoV-2 S VSV entry into Vero E6 cells stably expressing TMPRSS2 (Vero E6-TMPRSS2)[70] in a concentration-dependent manner with a half-maximal inhibitory

concentration (IC$_{50}$) of approximately 65 ng ml$^{-1}$ (Fig. 4a). No effect was observed using Vero E6 cells (lacking TMPRSS2), consistent with the fact that H1H7 specifically targets TMPRSS2 (ref. 39). Moreover, H1H7 at 5 µg ml$^{-1}$ markedly decreased infection of SARS-CoV-2 WA-1, BA.2, XBB.1.5 and CH.1.1 authentic viruses in Vero E6-TMPRSS2 but not in Vero E6 cells (Fig. 4b). These results suggest that H1H7 interferes with TMPRSS2-mediated entry at the plasma membrane but not with cathepsin-mediated, endosomal entry.

Although Omicron variants were suggested to bypass TMPRSS2 for cell entry[40,83,84], subsequent work indicated that TMPRSS2 remains essential for infection of Calu-3 human lung cells and of human airway and intestinal organoids[45,46,85], as well as for efficient replication in mouse lungs[86]. A recent study confirmed the key role of TMPRSS2 for replication of all SARS-CoV-2 variants and suggested that its apparent lack of involvement in some cell lines depends on whether ACE2 is interacting or not with tissue-specific solute carriers[87]. To assess the resilience of H1H7, we evaluated its ability to inhibit propagation of a panel of authentic SARS-CoV-2 variants in Vero E6 cells expressing TMPRSS2 jointly with ACE2 and the SLC6A19 solute carrier (TASL-19 cells)[87]. This cell line enables effective viral growth and syncytia formation of all SARS-CoV-2 variants in a TMPRSS2-dependent manner, as shown by nafamostat-mediated inhibition (Extended Data Fig. 6a–d). H1H7 inhibited propagation and syncytia formation of clade A (ancestral), Delta, BA.5, XBB.1.5, KP.3 and XFG isolates in a concentration-dependent manner, indicating retained efficacy against past and present SARS-CoV-2 variants (Fig. 4c).

Given that TMPRSS2 was proposed to activate membrane fusion for other coronaviruses[16,32,35–37,47], we evaluated the ability of H1H7 to inhibit entry of SARS-CoV-2, SARS-CoV-1 and MERS-CoV S VSV pseudoviruses into Calu-3 cells, a human lung cancer cell line with a predominant TMPRSS2-mediated entry route[39,88]. We observed that H1H7 inhibited all three pseudoviruses (Fig. 4d and Extended Data Fig. 6e–g), underscoring its broad-spectrum activity.

To elucidate the binding mode of H1H7 to TMPRSS2, we determined a 3.2-Å-resolution single-particle cryo-EM structure of a complex comprising the TMPRSS2-S441A ectodomain and the H1H7 Fab fragment (Fig. 4e, Extended Data Fig. 5c and Table 1). An anti-kappa light chain nanobody was also included to assist with structural determination by rigidifying the Fab and increasing the overall complex molecular mass[89,90]. The structure reveals that H1H7 binds to the SRCR domain, featuring contacts involving heavy and light chains through polar interactions and shape complementarity. The heavy-chain CDR2 and CDR3 loops dominate the paratope and jointly bury approximately 500 Å$^2$ at the interface with TMPRSS2 (Fig. 4e,f). H1H7 binds 40 Å away from the TMPRSS2 active site and did not affect the catalytic activity of the TMPRSS2 ectodomain toward a peptide substrate (Fig. 4g), suggesting that H1H7 inhibits SARS-CoV-2 without directly obstructing the active site. The TMPRSS2 orientation necessary for E-FIC S$_2'$ site cleavage indicates that binding of H1H7 to TMPRSS2 would cause a steric clash with the host membrane in which the fusion peptide is embedded, thereby preventing proteolytic processing (Fig. 4h).

## H1H7 protects mice against SARS-CoV-2 challenge

We next evaluated whether H1H7 could protect mice against SARS-CoV-2 challenge. H1H7 IgG was administered intraperitoneally to human TMPRSS2 knock-in C57BL/6 mice (hTMPRSS2-KI; wherein hTMPRSS2 is expressed by replacing the endogenous mouse TMPRSS2 locus) at doses of 25, 10 and 5 mg kg$^{-1}$ 1 day before intranasal challenge with 10$^5$ focus-forming units (FFU) of a SARS-CoV-2 B.1.351 isolate (Fig. 5a). H1H7 administration reduced levels of infectious virus and viral RNA in the lungs of treated animals at 4 and 6 days post infection (dpi) relative to mice receiving an isotype control antibody (Fig. 5b,c). Specifically, H1H7 mediated >100-fold and 1,000-fold respective reductions of lung infectious virus titers and viral RNA loads at 4 dpi when administered at 25 mg kg$^{-1}$, relative to the isotype control antibody. Furthermore, H1H7 promoted a reduction of

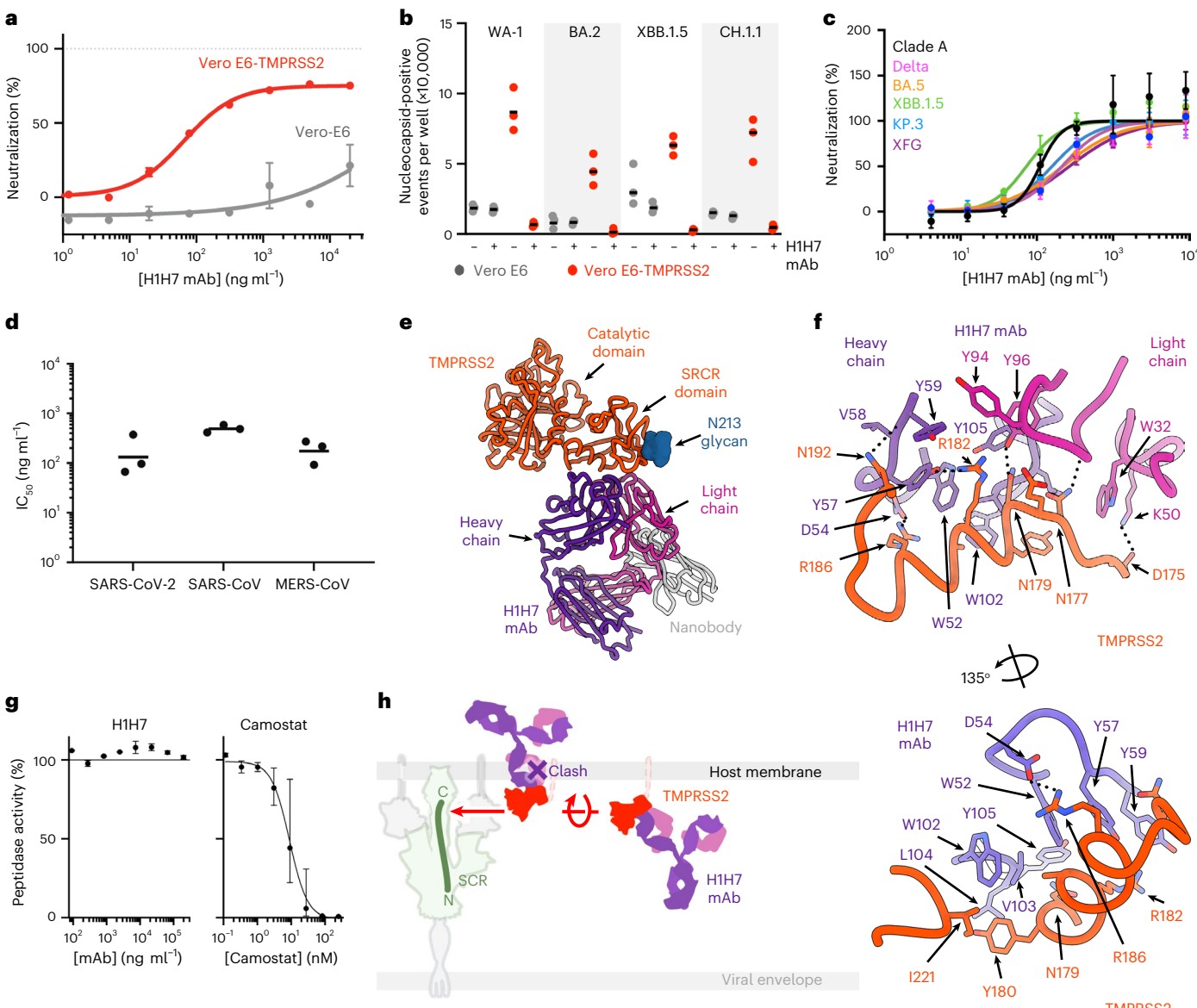

**Fig. 4 | The TMPRSS2-directed H1H7 antibody inhibits SARS-CoV-2.**
**a**, H1H7-mediated inhibition of SARS-CoV-2 Wuhan-Hu-1 S VSV pseudovirus entry into Vero E6-TMPRSS2 and Vero E6 cells. Data show the mean and s.e.m. for three technical replicates and are representative of two biological replicates. **b**, Infection of Vero E6-TMPRSS2 or Vero E6 cells by authentic SARS-CoV-2 WA-1, BA.2, XBB.1.5 and CH.1.1 viruses in the presence or absence of H1H7 at 5 μg ml⁻¹. Nucleocapsid-positive cells were counted after immunostaining with an anti-SARS-CoV-2 nucleocapsid antibody. Data show the mean and s.e.m. for three technical replicates and are representative of two biological replicates except for CH.1.1 for which a single biological replicate was performed. **c**, H1H7-mediated inhibition of infection of SARS-CoV-2 clade A (ancestral), Delta, BA.5, XBB.1.5, KP.3 and XFG authentic isolates in TASL-19 cells. Data represent the mean and s.d. of four technical replicates; data show one representative of three biological replicates. **d**, H1H7-mediated inhibition of SARS-CoV-2 Wuhan-Hu-1/G614, SARS-CoV-1 and MERS-CoV S VSV pseudovirus entry into Calu-3 cells. Each data point is a biological replicate (*n* = 3), with the geometric mean shown as lines. **e**, Cryo-EM structure of the H1H7 Fab in complex with the human TMPRSS2-S441A

ectodomain and a nanobody recognizing the Fab kappa light chain. The TMPRSS2 N213-linked glycan is shown as a blue surface. **f**, Close-up views of the interface between H1H7 and TMPRSS2. **g**, TMPRSS2 activity was measured using a fluorescent peptide substrate (Dabcyl-RRARSVASQSI-(Glu)EDANS) and 0.00185 mg ml⁻¹ TMPRSS2 in the presence of a threefold dilution series of H1H7 mAb (starting at 0.2 mg ml⁻¹; left) or camostat (starting at 250 nM; right). Data represent the mean and s.d. of two technical replicates. **h**, Schematic model (to scale) showing SARS-CoV-2 S E-FIC bound to ACE2 (transparent), alongside membrane-anchored TMPRSS2 (red) bound to the H1H7 antibody (purple and pink). The red arrow indicates the trajectory from the TMPRSS2 active site to the S₂′ cleavage site. TMPRSS2 is shown in two orientations: positioned to access the E-FIC S₂′ site in which the target peptide extends from the N to C terminus toward the host membrane (left) and in an orientation compatible with H1H7 binding (right). H1H7 engagement of TMPRSS2 would be sterically incompatible with the orientation needed to access the S₂′ cleavage site because of the presence of the host membrane.

viral RNA load in nasal turbinates at 4 and 6 dpi relative to the control group (Fig. 5d,e).

Although TMPRSS2 was previously reported to be involved in SARS-CoV-1 infection and pathology[35–37,49], H1H7 administration did not impact viral loads or survival after SARS-CoV-MA15 infection of

hTMPRSS2-KI mice (Extended Data Fig. 7). These results suggest the use of another host protease in this animal model (possibly because of redundancy), concurring with prior work reporting retained SARS-CoV-1 viral spread in TMPRSS2-deficient mice (albeit reduced compared to wild-type mice)[37].

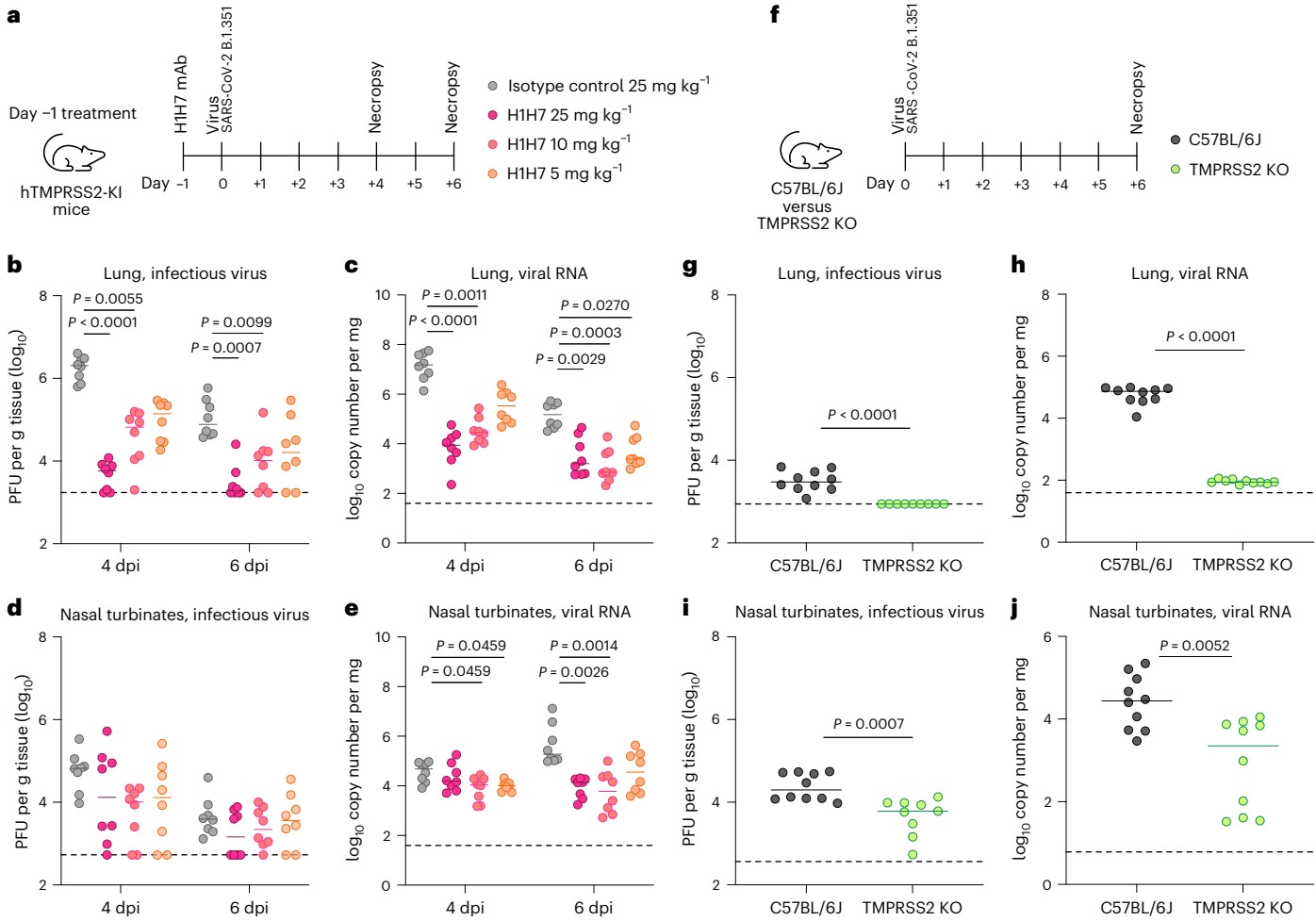

**Fig. 5 | The H1H7 antibody inhibits SARS-CoV-2 infection by targeting the host TMPRSS2 protease. a**, Study design. Male and female hTMPRSS2-KI mice (8 weeks old) were administered 5, 10 or 25 mg kg⁻¹ of H1H7 mAb or 25 mg kg⁻¹ isotype control 1 day before intranasal inoculation with $10^5$ FFU of SARS-CoV-2 B.1.351. **b–e**, Lung infectious viral titers (**b**), lung viral RNA loads (**c**), nasal turbinate infectious viral titers (**d**) and nasal turbinate viral RNA loads (**e**) were evaluated at 4 and 6 dpi. **f**, Study design. Female wild-type C57BL/6J or TMPRSS2-KO mice (8 weeks old) were inoculated intranasally with $10^5$ FFU of

SARS-CoV-2 B.1.351. **g–i**, Lung infectious viral titers (**g**), lung viral RNA loads (**h**), nasal turbinate infectious viral titers (**i**) and nasal turbinate viral RNA loads (**j**) were evaluated at 6 dpi. Median values are shown as bars. Statistical analysis was performed using a Kruskal–Wallis analysis of variance with Dunn's post hoc test (**b–e**) or two-tailed Mann–Whitney test (**g–j**). For each graph, dotted lines indicate the limit of detection. In **b–e**, $n = 8$ for each graph. In **g–j**, $n = 10$ (left) and 9 (right) (**g**), $n = 10$ (**h**), $n = 10$ (left) and 9 (right) (**i**) and $n = 10$ (**j**) for each graph.

Given that H1H7 mediated superior protection of the lungs than the nasal turbinates, we set out to determine the contribution of TMPRSS2 to viral replication in the lower and the upper airways. Intranasal infection of wild-type C57BL/6 mice or TMPRSS2-knockout (TMPRSS2-KO) mice with $10^5$ FFU of SARS-CoV-2 B.1.351 revealed that loss of TMPRSS2 had a greater effect on levels of infectious virus and viral RNA in the lungs than in the nasal turbinates at 6 dpi (Fig. 5f–j), consistent with prior data with mouse TMPRSS2 (ref. 91). Collectively, these data show that prophylactic treatment with 25 mg kg⁻¹ H1H7 reduced viral burden comparably to genetic KO of TMPRSS2, underscoring that TMPRSS2-blocking antibodies may be an attractive alternative to direct viral targeting.

## Discussion

A mechanistic understanding of TMPRSS2-mediated cleavage in the coronavirus S transition from prefusion to postfusion has remained elusive. We show that TMPRSS2 recognizes and cleaves E-FIC at the R815 S₂' site, promoting shedding of ACE2-bound S₁ and S₂ refolding to the postfusion conformation. This discovery resolves two conundrums: (1) the coronavirus S₂' cleavage site is inaccessible in prefusion S and (2) TMPRSS2 does not protrude enough from the host membrane

to reach the S₂' cleavage site in prefusion S bound to ACE2. In contrast, the S₂' cleavage site is accessible in E-FIC and close to the host membrane[59–61,63]. Consequently, E-FIC but not prefusion S is compatible with efficient TMPRSS2 binding and cleavage at S₂', explaining the marked ACE2-induced binding enhancement of SARS-CoV-2 neutralizing antibodies to this (formerly named fusion peptide[24]) site[19,20]. This multistep transition from the prefusion to the postfusion S conformation and the requirement for ACE2 and TMPRSS2 to be in close proximity in the target membrane likely ensure the coordination of multiple S trimers required for formation of a fusion pore[26].

Our results suggest that stabilization of E-FIC, instead of prefusion S₂, might be necessary for efficient vaccine elicitation of S₂'-directed antibodies[71,92–94], given that it is the former conformation that is recognized by these broad inhibitors. The stabilized E-FIC$_{S-v3}$ design will, therefore, be a valuable tool for future immunological and biochemical studies. We note that the conformation of the residues surrounding the S₂' cleavage site is distinct for apo E-FIC (largely disordered) and for E-FIC bound to S₂'-directed antibodies (helical), suggesting selection of a specific conformation[19–21]. Our findings reveal the molecular mechanism of S₂' site-directed broad antibody neutralization and open new avenues to target the TMPRSS2-mediated coronavirus activation step.

Several coronaviruses can enter cells using alternative proteases to TMPRSS2 or through fusion with the endolysosomal membranes in vitro, which may be associated with different requirements and residue specificity for cleavage[14,18,32,36,41,52,53,95,96]. For instance, cathepsin L can cleave prefusion SARS-CoV-2 S at noncanonical sites[97], possibly explaining the mutational tolerance at residue 815 observed in cell lines that do not mimic important aspects of entry in relevant respiratory epithelial cells[31]. The enhanced coronavirus entry into cells, replication kinetics and S-mediated fusion observed in the presence of TMPRSS2 and other proteases at the target cell surface, relative to the endosomal route, indicate that plasma membrane entry is favored[11,35,41,47,75–78,87,98]. Accordingly, passaging SARS-CoV-2 and HCoV-229E in the presence of TMPRSS2 alleviates the fitness advantage of cell-culture-adapted isolates[54,56,99] and renders SARS-CoV-2 insensitive to chloroquine-mediated inhibition of endosomal entry; the latter compound did not prove a suitable COVID-19 countermeasure[100]. These results highlight the key role of E-FIC $S_2'$ cleavage in S-mediated entry and fusion, explaining the broad-spectrum inhibition and protection mediated by $S_2'$ site-directed antibodies[19–21], small-molecule TMPRSS2 inhibitors[42,49,62,64,101] and the TMPRSS2-directed H1H7 antibody.

Given that H1H7 does not interfere with the catalytic activity of TMPRSS2 directly, we propose that it functions through steric hindrance; H1H7 binding to TMPRSS2 would sterically hinder access to the $S_2'$ cleavage site because of the presence of the target cell membrane and neighboring ACE2-bound RBDs in E-FIC. By targeting a key host determinant, H1H7 broadly inhibits a panel of SARS-CoV-2 variants spanning the entire duration of the COVID-19 pandemic up to this day and protects human TMPRSS2-KI mice from SARS-CoV-2 challenge. Moreover, H1H7 blocks entry of SARS-CoV-1 and MERS-CoV pseudoviruses into cells, further emphasizing the importance of TMPRSS2-mediated coronavirus activation for cell entry. The lack of protection against SARS-CoV-1 MA15 in mice, however, warrants further exploration to determine whether the process of mouse adaptation of the challenge virus influenced TMPRSS2 usage[102], given that camostat protected mice against a distinct mouse-adapted SARS-CoV-1 virus[49].

Although the physiological functions of TMPRSS2 remain incompletely defined, several lines of evidence indicate that H1H7 is unlikely to elicit adverse effects. TMPRSS2-KO mice display no developmental abnormalities[103] and the serine protease inhibitor camostat mesylate (active against TMPRSS2) is clinically approved[39]. Moreover, H1H7 does not inhibit the TMPRSS2 catalytic activity in vitro and administration of a high dose (25 mg kg⁻¹) of H1H7 in mice was well tolerated. Unlike small-molecule inhibitors such as camostat, which suffers from low specificity and a short half-life, antibodies such as H1H7 can be engineered with Fc modifications to improve bioavailability and durability—key factors for clinical efficacy[42,104].

This approach may limit the emergence of escape mutants and mitigate the impact of antigenic changes, which have reduced the efficacy of monoclonal antibodies, small-molecule inhibitors and vaccines targeting the virus directly[105–109]. Although the contribution of TMPRSS2 to the entry of SARS-CoV-2 Omicron variants has been debated[40,84], accumulating evidence indicates that it continues to have a key role in infection[45,86,91], underscoring the ongoing relevance of TMPRSS2-targeted interventions, as shown here using TASL-19 cells[87]. This host-directed strategy may, therefore, offer broader protection against SARS-CoV-2 variants and other pathogens using TMPRSS2 to activate their fusion protein, including coronaviruses, influenza viruses, parainfluenzaviruses and human metapneumovirus.

## Online content

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

[1]Department of Biochemistry, University of Washington, Seattle, WA, USA. [2]Department of Medicine, Washington University School of Medicine, St. Louis, MO, USA. [3]Department of Molecular Microbiology, Washington University School of Medicine, St. Louis, MO, USA. [4]Howard Hughes Medical Institute, Seattle, WA, USA. [5]Department of Chemistry, University of Washington, Seattle, WA, USA. [6]The Kirby Institute, The University of New South Wales, Sydney, New South Wales, Australia. [7]Vir Biotechnology, San Francisco, CA, USA. [8]Institute for Research in Biomedicine, Università della Svizzera Italiana, Bellinzona, Switzerland. [9]Institute of Microbiology, ETH Zürich, Zurich, Switzerland. [10]Bioqual, Rockville, MD, USA. [11]Vir Biotechnology, Bellinzona, Switzerland. [12]National Institute of Molecular Genetics, Milano, Italy. [13]Department of Pathology and Immunology, Washington University School of Medicine, St. Louis, MO, USA. [14]Andrew M. and Jane M. Bursky Center for Human Immunology and Immunotherapy Programs, Washington University School of Medicine, St. Louis, MO, USA. [15]Present address: Department of Laboratory Medicine & Pathobiology, University of Toronto, Toronto, Ontario, Canada. [16]Present address: A*STAR Infectious Diseases Labs (A*STAR IDL), Agency for Science, Technology and Research (A*STAR), Singapore, Singapore. [17]Present address: Third Rock Ventures, Boston, MA, USA. [18]These authors contributed equally: Matthew McCallum, James Brett Case, Jack T. Brown, Young-Jun Park. ✉e-mail: mdiamond@wustl.edu; dveesler@uw.edu

## Methods

The work described in this manuscript complies with all relevant ethical regulations. Animal studies were carried out in accordance with the recommendations in the Guide for the Care and Use of Laboratory Animals of the National Institutes of Health (NIH). The protocols were approved by the Institutional Animal Care and Use Committee (IACUC) at the Washington University School of Medicine (assurance no. A3381-01) for SARS-CoV-2 challenge and by the IACUC at BIOQUAL (protocol no. 23-116P) for SARS-CoV-1 challenge.

### Cells

Vero E6 cells (CRL-1586, American Type Culture Collection (ATCC)) were maintained in DMEM (Invitrogen) supplemented with 10% FBS (Omega Scientific) and 100 U per ml penicillin–streptomycin (P/S) (Invitrogen). Vero cells expressing TMPRSS2 (ref. 70) or hACE2-TMPRSS2 (a gift from A. Creanga and B. Graham, NIH) were maintained as Vero E6 cells, with the addition of 5 µg ml$^{-1}$ blasticidin (Vero E6-TMPRSS2) or 10 µg ml$^{-1}$ puromycin (Vero E6-hACE2-TMPRSS2). HEK293T (ATCC, CRL-3216) cells were cultured in DMEM (Gibco) supplemented with 10% FBS (Cytiva) and 1% P/S (Life Tech). Calu-3 cells (ATCC, HTB-55) were maintained in DMEM/F-12 GlutaMAX supplement (Gibco) supplemented with 10% FBS (Cytiva). All cell lines were maintained at 37 °C with 5% CO$_2$. TASL-19 cells were previously described[87].

### Construct design

The SARS-CoV-2 S$_{ecto}$ constructs were synthesized and cloned into pCMV (R815) and pcDNA3.4 (R815H) vectors. They comprise an N-terminal signal peptide (MGILPSPGMPALLSLVSLLSVLLMGC-VAETGT), S residues 23–1220 with the D614G substitution and the polybasic S$_1$/S$_2$ cleavage site (R$_{682}$RAR$_{685}$) substituted to S$_{682}$GAR$_{685}$, a C-terminal TEV protease cleavage site, a GGGGSG linker, a C-terminal T4 foldon trimerization domain and an octahistidine tag.

The monomeric human ACE2 (peptidase) ectodomain (residues 19–615) was designed with an N-terminal secretion signal peptide (MGSLQPLATLYLLGMLVASVLA) and a C-terminal decahistidine tag and was synthesized into the pTWIST CMV expression vector.

Human TMPRSS2 ectodomain constructs were designed as previously described[60]. The inactive TMPRSS2 construct comprises an azurocidin signal peptide (MTRLTVLALLAGLLASSRA), a serine residue, a SUMO tag (UniProt Q12306 ; 1–94, MSDSEVNQEAKPEVK-PEVKPETHINLKVSDGSSEIFFKIKKTTPLRRLMEAFAKRQGKEMDSLRFLY-DGIRIQADQTPEDLDMEDNDIIEAHREN), an enteropeptidase cleavage site (NDDDDK), a TT linker, residues 109–492 of TMPRSS2 (UniProt O15393; numbering without start methionine), a C-terminal enteropeptidase site (DDDDK), an SG linker and an octahistidine tag. TMPRSS2 harbors the following substitutions: an internal enteropeptidase site introduced by substituting R$_{251}$QSR$_{254}$ to DDDDK, the S441A substitution to make it catalytically inactive and the T447C stabilizing substitution. The active TMPRSS2 ectodomain includes a C-terminal flexible SGSG linker and an octahistidine tag for biochemical characterization.

The NL63-S$_2$′/HKU1-RBD construct was generated by In-Fusion mutagenesis (Takara, 638949) of a previously described HKU1-RBD construct[60]. It comprises a signal peptide (MGILPSPGMPALLSLVS-LLSVLLMGCVAETGT), HKU1 (UniProt Q5MQD0; isolate N1) residues 320–509, HCoV-NL63 residues I867–L877 (IAGRSALEDLL; UniProt Q6Q1S2), a GSGSGSP linker and HKU1 (UniProt Q5MQD0; isolate N1) residues 511–614, a GSSGGS linker, an Avi tag (GLNDIFEAQKIEWHE), a GGS linker and an octahistidine tag.

The heavy and light (kappa) chain sequences for the H1H7 mAb were obtained from US patent US20190300625A1. Codon-optimized genes were synthesized by GenScript and individually cloned into pCDNA3.1(+) vectors, each with an N-terminal CD5 secretion signal (MPMGSLQPLATLYLLGMLVASVLA).

Nirsevimab variable region sequences (VH and VL) were retrieved from the IMGT database (ID853). H1H7 variable sequences were retrieved from US patent US20190300625A1. All sequences were codon-optimized for hamster cell expression, synthesized by GenScript, and subcloned into IgG1 expression plasmids.

The SARS-CoV-2 HexaPro S ectodomain (residues 1–1208) construct contains the HexaPro substitutions[80] (F817P, A892P, A899P, A942P, K986P and V987P), the R682G, R683S and R685S substitutions of the S$_1$/S$_2$ cleavage site, a C-terminal linker, a foldon, HRV 3C site (LEVLFQGP), another linker, an Avi tag and a final short linker before an octahistidine tag in a pcDNA3.1(-) plasmid.

The SARS-CoV-2 E-FIC$_{S-v1}$ construct was designed to stabilize the E-FIC conformation and it encodes an azurocidin signal peptide, a serine residue, a SUMO tag (UniProt Q12306; 1–94), an enterokinase cleavage site, a TGTGS linker, HIV-1 gp41 residues 538–578 (corresponding to HR1, TVQVRKLLSGIVQQQSNLLRAIEAQQHLLKLTVWGIKQLQA), a single oP3h repeat[79] (RINAIET) preceded by four residues (QVEE) to match the HR1 heptad repeat, an alanine–isoleucine linker, SARS-CoV-2 S residues 914–1131 (corresponding roughly to the HR1-CH-βH-CD region, with the D985A substitution), a threonine residue, SARS-CoV-2 S residues 703–828 (corresponding roughly to the β-UH-SCR region), a short linker (PDPGSNEE), HIV-1 gp41 residues 628–665 (corresponding to HR2, WLQWDKEISNYTHIIYELIEESQKQQEKNEQELLELDK), an SSGGT linker, a second enterokinase cleavage site, a modified Avi tag (TGSLNDIFEAQK-IEWHE), a QGS linker and a C-terminal octahistidine tag cloned into a pCMV expression plasmid (MTRLTVLALLAGLLASSRASMSDSEVN-QEAKPEVKPEVKPETHINLKVSDGSSEIFFKIKKTTPLRRLMEAFAKRQG-KEMDSLRFLYDGIRIQADQTPEDLDMEDNDIIEAHRENDDDDKTGTG-STVQVRKLLSGIVQQQSNLLRAIEAQQHLLKLTVWGIKQLQAQVEERI-NAIETAINVLYENQKLIANQFNSAIGKIQDSLSSTASALGKLQDVVN-QNAQALNTLVKQLSSNFGAISSVLNDILSRLAKVEAEVQIDRLITGRLQS-LQTYVTQQLIRAAEIRASANLAATKMSECVLGQSKRVDFCGKGYHLMSF-PQSAPHGVVFLHVTYVPAQEKNFTTAPAICHDGKAHFPREGVFVSNGTH-WFVTQRNFYEPQIITTDNTFVSGNCDVVIGTNSVAYSNNSIAIPTNFTISVT-TEILPVSMTKTSVDCTMYICGDSTECSNLLLQYGSFCTQLNRALTGIA-VEQDKNTQEVFAQVKQIYKTPPIKDFGGFNFSQILPDPSKPSKRSFIEDLL-FNKVTLPDPGSNEEWLQWDKEISNYTHIIYELIEESQKQQEKNEQEL-LELDKSSGGTDDDDKTGSLNDIFEAQKIEWHEQGSHHHHHHHH). In silico design began by testing whether the HR2 region of SARS-CoV-2 S could be substituted by the β-UH segment through circular permutation and removing the aggregation-prone FPR motif while preserving the E-FIC-like conformation of the HR1 and S$_2$′ regions. Iterative modeling and evaluation with AlphaFold3 (ref. 110) was used to assess folding and interdomain organization across designs. Initial predictions indicated that the S$_2$′ segment remained disordered in the absence of additional structural constraints. To stabilize this region, the HR1 and HR2 regions of HIV-1 gp41 were introduced to clamp the S$_2$′ segment in place, connecting gp41 HR1 to SARS-CoV-2 HR1 through a modified oP3h repeat (QVEERINAIETAI). A short structured linker (PDPGSNEE) connecting SARS-CoV-2 S SCR to gp41 HR2 was designed and optimized using RFdiffusion[111], ProteinMPNN[112] and AlphaFold3 to maximize predicted stability and minimize interdomain strain. The final construct was selected on the basis of high predicted local distance difference test confidence scores and maintenance of the intended domain topology. E-FIC$_{S-v2}$ is identical to E-FIC$_{S-v1}$ except that five additional SCR C-terminal residues were included (extending the construct to residues 703–833) and the subsequent short linker (TGTVGPEE) was optimized with ProteinMPNN[112] using the experimentally determined E-FIC$_{S-v1}$ structure and AlphaFold3 to yield E-FIC$_{S-v2}$ (MTRLTVLALLAGLLASSRASMSDSEVNQEAKPEVKPEVKPETHINLKVS-DGSSEIFFKIKKTTPLRRLMEAFAKRQGKEMDSLRFLYDGIRIQADQT-PEDLDMEDNDIIEAHRENDDDDKTGTGSTVQVRKLLSGIVQQQSNLL-RAIEAQQHLLKLTVWGIKQLQAQVEERINAIETAINVLYENQKLIANQFN-SAIGKIQDSLSSTASALGKLQDVVNQNAQALNTLVKQLSSNFGAISS-VLNDILSRLAKVEAEVQIDRLITGRLQSLQTYVTQQLIRAAEIRASANLAATK-MSECVLGQSKRVDFCGKGYHLMSFPQSAPHGVVFLHVTYVPAQEKNFT-TAPAICHDGKAHFPREGVFVSNGTHWFVTQRNFYEPQIITTDNTFVS-

GNCDVVIGTNSVAYSNNSIAIPTNFTISVTTEILPVSMTKTSVDCTMY-
ICGDSTECSNLLLQYGSFCTQLNRALTGIAVEQDKNTQEVFAQVKQIYK-
TPPIKDFGGFNFSQILPDPSKPSKRSFIEDLLFNKVTLADAGFTGTVGPEE-
WLQWDKEISNYTHIIYELIEESQKQQEKNEQELLELDKSSGGTDDDD-
KTGSLNDIFEAQKIEWHEQGSHHHHHHHH). E-FIC$_{S-v3}$ is identical to
E-FIC$_{S-v2}$ except that gp41 was replaced with a sequence-optimized
version of equine infectious anemia virus gp45, termed gp45m. More-
over, the oP3h and alanine–isoleucine linker were replaced with an
alternative optimized sequence (EVEAIQKAI) and the short linker
between SARS-CoV-2 S SCR and gp45m HR2 was replaced with an
alternative sequence (TIDKEK). The gp45 glycoprotein was selected
as an alternative to gp41 as it is structurally similar to gp41 (ref. [113])
despite low sequence identity, thereby minimizing the potential for
cross-reactivity with HIV-1 antibodies. The gp45 sequence was opti-
mized using ProteinMPNN and AlphaFold3 to generate gp45m, which
shares 49% sequence identity with gp45 and 25% identity with gp41.
The short structured linkers connecting gp45m HR1 to the SARS-CoV-2
HR1 (EVEAIQKAI) and the SARS-CoV-2 S SCR to gp45m HR2 (TIDKEK)
were designed and optimized using RFdiffusion[111], ProteinMPNN[112] and
AlphaFold3 to maximize predicted stability and minimize interdomain
strain in E-FIC$_{S-v3}$ (MTRLTVLALLLAGLLASSRASMSDSEVNQEAKPEVK-
PEVKPETHINLKVSDGSSEIFFKIKKTTPLRRLMEAFAKRQGKEMDSLR-
FLYDGIRIQADQTPEDLDMEDNDIIEAHRENDDDDKTGTGSQSNAES-
VQNHTFEVLNNTIRALELILRKLEILYEMILQLHEEVEAIQKAINVLYEN-
QKLIANQFNSAIGKIQDSLSSTASALGKLQDVVNQNAQALNTLVKQLSS-
NFGAISSVLNDILSRLAKVEAEVQIDRLITGRLQSLQTYVTQQLIRAAEI-
RASANLAATKMSECVLGQSKRVDFCGKGYHLMSFPQSAPHGVVFLHV-
TYVPAQEKNFTTAPAICHDGKAHFPREGVFVSNGTHWFVTQRNFYEP-
QIITTDNTFVSGNCDVVIGTNSVAYSNNSIAIPTNFTISVTTEILPVSMTK-
TSVDCTMYICGDSTECSNLLLQYGSFCTQLNRALTGIAVEQDKNTQEV-
FAQVKQIYKTPPIKDFGGFNFSQILPDPSKPSKRSFIEDLLFNKVTLADAG-
FTIDKEKWKEWRKEMENLTKEIKETLEEARKTLKQAEETFKTPSSGGTD-
DDDKTGSLNDIFEAQKIEWHEQGSHHHHHHHH).

The anti-kappa nanobody[89] was cloned into pET23a with an
N-terminal DsbA signal peptide for periplasmic expression and
C-terminal octahistidine tag.

## Recombinant protein expression and purification

For negative-stain EM, SDS–PAGE and western blot experiments,
SARS-CoV-2 S ectodomains (including SARS-CoV-2 E-FIC$_S$), human
ACE2 ectodomain and human TMPRSS2 were expressed in Expi293 cells
(Thermo) at 37 °C and 8% CO$_2$ with constant rotation at 130 rpm. Cells
were transfected using the Expifectamine293 transfection kit (Thermo)
in accordance with the manufacturer's protocol. Then, 4–5 days after
transfection, Expi293 cell supernatant was clarified by centrifugation
at 4,121$g$ for 30 min, supplemented with 25 mM phosphate pH 8.0 and
300 mM NaCl. Supernatant was then bound to Ni Excel resin (Cytiva)
previously equilibrated in 25 mM phosphate pH 8.0 and 300 mM NaCl.
Nickel resins were washed with 20–40 column volumes (CVs) of 25 mM
phosphate pH 8.0, 300 mM NaCl and 20–40 mM imidazole for S ecto-
domains or 20 mM imidazole for ACE2 and TMPRSS2 ectodomains.
TMPRSS2 ectodomains used for cryo-EM were purified using a similar
protocol, with the following modifications: Tris-HCl buffer pH 7.5 was
used in place of phosphate buffer pH 8.0 and TALON resin (Takara) was
substituted for nickel affinity resin. Protein was eluted using 25 mM
phosphate pH 8.0, 300 mM NaCl and 300 mM imidazole before being
buffer-exchanged to 50 mM Tris-HCl pH 8.0, 150 mM NaCl and con-
centrated using a centrifugal filter device (Amicon or Corning) with
a molecular weight cutoff (MWCO) of 100 kDa for S ectodomains and
30 kDa for ACE2 and TMPRSS2 ectodomains. Inactive TMPRSS2 was
diluted to 0.3–0.5 mg ml$^{-1}$ in 50 mM Tris pH 7.4, 150 mM NaCl and 10 ml
of CaCl$_2$ and was digested with 1:500–1:700 (v/v) EKMax Enterokinase
(Invitrogen) overnight at room temperature. S ectodomains were fur-
ther purified on a Superose 6 increase 10/300 size-exclusion column
equilibrated in 50 mM Tris pH 7.4 and 150 mM NaCl. ACE2 and TMPRSS2

ectodomains were further purified on a Superdex 200 Increase 10/300
size-exclusion column (Cytiva) equilibrated in 50 mM Tris pH 7.4 and
150 mM NaCl. Fractions containing monodisperse protein were then
flash-frozen and stored at −80 °C until use.

For BLI experiments with TMPRSS2 and H1H7 or cryo-EM sample
preparation, biotinylated inactive human TMPRSS2 and the H1H7 mAb
were expressed in Expi293 cells (ThermoFisher Scientific) and cultured
at 37 °C in a humidified 8% CO$_2$ incubator with constant rotation at
130 rpm. Cells were cultured and transfected as outlined by the manu-
facturer using the Expifectamine 293 transfection kit (Thermo Fisher
Scientific) and cultured for 4 days before harvest. Cell culture superna-
tants were clarified using centrifugation and harvested using a HisTrap
HP nickel Sepharose column (Cytiva) for TMPRSS2 and HiTrap protein
A HP column (Cytiva) for the H1H7 mAb. The columns were washed
with 20 CVs of buffer containing 25 mM Tris, 150 mM NaCl and 10 mM
imidazole pH 8.0 for the TMPRSS2 column or 20 mM sodium phosphate
pH 8.0 for the H1H7 column. Afterwards, the TMPRSS2 column was
eluted using 15 CVs of 25 mM Tris, 150 mM NaCl and 300 mM imidazole
pH 8.0 and the H1H7 column was eluted using 10 CVs of 0.1 M citric acid
pH 3.0 directly into 1 M Tris-HCl pH 9.0 to immediately neutralize the
pH to 8.0. TMPRSS2 and H1H7 mAb were buffer-exchanged into 25 mM
Tris and 150 mM NaCl pH 8.0 using Amicon Ultra-15 centrifugal filter
units, 30-kDa MWCO for TMPRSS2 and 100-kDa MWCO for H1H7. The
inactive TMPRSS2 was diluted to a concentration of 0.3 mg ml$^{-1}$ in the
same buffer and supplemented with a 1:10 (v/v) addition of 10× EK
digestion buffer (500 mM Tris-HCl, 10 mM CaCl$_2$ and 1% Tween-20
pH 8.0) and a 1:700 (v/v) addition of EKMax Enterokinase (Invitrogen).
The TMPRSS2 digestion mix was incubated overnight at room tem-
perature to cleave the TMPRSS2 construct before concentration using
Amicon Ultra-15 centrifugal filter unit (10-kDa MWCO) and biotinylated
at a final concentration of roughly 40 mM TMPRSS2 overnight at 4 °C
using the reagents and components of the Avidity BirA ligase protocol
at the manufacturer's recommended dilutions. For further purification,
the digested and biotinylated inactive TMPRSS2 was run through a
Superdex 200 Increase 10/300 column (Cytiva) equilibrated in 25 mM
Tris, 150 mM NaCl pH 8.0, before flash-freezing using liquid nitrogen
and stored at −80 °C until later use.

To generate H1H7 Fab fragments, H1H7 IgG was digested with endo-
proteinase LysC (New England Biolabs) at an enzyme-to-substrate ratio
of 1:100 or 1:500 (w/w) for 18 h at 37 °C. Fc fragments were removed
by affinity chromatography using a HiTrap protein A column (Cytiva)
and the flowthrough containing Fab fragments was further purified
by size-exclusion chromatography (SEC) on a Superdex 75 Increase
10/300 GL column (Cytiva) equilibrated in 50 mM HEPES pH 7.5 and
150 mM NaCl. For cryo-EM samples, SEC purification was replaced with
filtration through an Amicon Ultra-0.5 centrifugal filter unit (100-kDa
MWCO; Millipore), retaining the flowthrough to remove uncleaved IgG
and aggregates. Purified H1H7 Fab was flash-frozen in liquid nitrogen
and stored at −80 °C until use.

For BLI experiments with TMPRSS2 and coronavirus S$_2$' peptides,
as well as for cryo-EM sample preparation, biotinylated NL63-S$_2$'/
HKU1-RBD and inactive TMPRSS2 were expressed in Expi293 cells
(Thermo) at 37 °C and 8% CO$_2$ with constant rotation at 130 rpm. Cells
were transfected using the Expifectamine 293 transfection kit (Thermo)
in accordance with the manufacturer's protocol. Then, 5 days after
transfection, Expi293 cell supernatant was clarified by centrifugation
at 2,739$g$ for 20 min and supplemented with 25 mM Tris-HCl pH 8.0,
150 mM NaCl, 1 mM imidazole and 0.7 mM CoCl$_2$. Supernatant was
then bound to TALON metal affinity resin (Takara) previously equili-
brated in 25 mM Tris-HCl and 150 mM NaCl pH 8.0. TALON resins were
washed with 600 ml of 25 mM Tris-HCl pH 8.0, 150 mM NaCl and 10 mM
imidazole. The purified proteins were eluted with 25 mM Tris-HCl
pH 8.0, 150 mM NaCl and 300 mM Imidazole. The eluted NL63-S$_2$'/
HKU1-RBD was buffer-exchanged with 50 mM Tris-HCl pH 8.0, 150 mM
NaCl and biotinylated at 40 μM overnight at 4 °C using the reagents and

components of the Avidity BirA ligase protocol at the manufacturer's recommended dilutions. The inactive TMPRSS2 was buffer-exchanged with 50 mM Tris-HCl pH 8.0, 150 mM NaCl and cleaved according to the aforementioned EKMax digestion protocol. The biotinylated NL63-S$_2$′/HKU1-RBD and digested inactive TMPRSS2 were further purified on a Superdex 200 Increase 10/300 GL size-exclusion column (Cytiva) equilibrated in 50 mM Tris-HCl pH 8.0 and 150 mM NaCl.

The SARS-CoV-2 HexaPro S construct used for BLI experiments was expressed in Expi293 cells (Thermo) at 37 °C and 8% CO$_2$ with constant rotation at 130 rpm. Cells were transfected using the Expifectamine 293 transfection kit (Thermo) and cultured for 4 days before harvest. The cell culture supernatant was clarified by centrifugation and harvested using a HisTrap HP (Cytiva) nickel Sepharose column. The column was washed with 20 CVs of 25 mM sodium phosphate, 300 mM NaCl and 10 mM imidazole pH 8.0 and the protein was eluted with 10 CVs of 25 mM sodium phosphate, 300 mM NaCl and 500 mM imidazole pH 8.0. The protein was then buffer-exchanged into 20 mM sodium phosphate and 100 mM NaCl pH 8.0 using Amicon Ultra-15 centrifugal filter units (100-kDa MWCO), flash-frozen using liquid nitrogen and stored at −80 °C for future analysis.

For in vitro neutralization and in vivo experiments, ExpiCHO cells were transiently transfected with heavy-chain and light-chain expression vectors for H1H7 and nirsevimab in a 1:1 ratio using the ExpiFectamine CHO transfection kit (Thermo Fisher Scientific). After 8 days of expression, mAbs were purified from the supernatant by protein A affinity chromatography.

For expression and purification of monoclonal antibodies, VN01H1, C13A7, C13B8, C13C9, C28F8, C77G12 and VP12E7 IgG, Expi293 (Gibco) cells were transiently transfected with heavy-chain and light-chain expression vectors, as previously described[114]. Affinity purification was performed on ÄKTA Pure 25 (Cytiva) operated by UNICORN 6.4, using HiTrap protein A columns (Cytiva, GE17-5079-01). Buffer exchange to PBS was performed with a HiPrep 26/10 desalting column (Cytiva, GE17-5087-01). The final products were sterilized by filtration through 0.22-μm filters and stored at 4 °C.

For cryo-EM of the VN01H1 Fab-bound E-FIC–hACE2 complex, the VN01H1 IgG was fragmented using LysC digestion in 1:500 mass ratio for 6 h at 37 °C and the Fab fragments were purified using a HiTrap protein A affinity column. The HiTrap protein A column was equilibrated with a binding buffer containing 20 mM sodium phosphate pH 7.4 before injecting the solution of LysC-cleaved Fab fragments. The collected flowthrough fractions were buffer-exchanged to PBS using Amicon Ultra-15 centrifugal filter units (30-kDa MWCO).

For cryo-EM sample preparation, the anti-kappa nanobody was expressed and purified from the periplasm of Escherichia coli BL21(DE3). A 1-L culture was grown in Luria–Bertani medium at 37 °C to an optical density (A$_{600}$) of 0.6, induced with 1 mM IPTG and incubated overnight at 18 °C with shaking. Cells were harvested by centrifugation and stored at −80 °C until purification. For periplasmic extraction, the cell pellet was thawed and resuspended in 40 ml of extraction buffer containing 20% (w/v) sucrose, 50 mM HEPES pH 7.5, 0.5 mM EDTA, 1 mM PMSF and 1% NP-40 (IGEPAL). The suspension was rocked for 30 min at room temperature to permeabilize the outer membrane and release periplasmic contents, followed by clarification through centrifugation at 7,000g for 15 min at 4 °C. The clarified lysate was incubated with 5 ml of TALON cobalt–NTA resin (Takara Bio) for 30 min under gentle agitation to capture the His-tagged nanobody. The resin was transferred to a gravity-flow column, washed with 500 ml of wash buffer (50 mM HEPES pH 7.5, 150 mM NaCl and 5 mM imidazole) and eluted with elution buffer containing 600 mM imidazole. The eluate was concentrated using a centrifugal concentrator (10-kDa MWCO) and further purified by SEC on a Superdex 75 Increase 10/300 GL column equilibrated in 50 mM HEPES pH 7.5 and 150 mM NaCl. The purified nanobody was concentrated to the desired working concentration, flash-frozen in liquid nitrogen and stored at −80 °C until use in cryo-EM grid preparation.

For cryo-EM analysis of the H1H7 Fab bound by the anti-kappa nanobody and inactive (S441A) TMPRSS2, purified components were combined in a total reaction volume of 573 μl, consisting of 33 μl of 5.6 mg ml$^{-1}$ anti-kappa nanobody, 40 μl of 9 mg ml$^{-1}$ H1H7 Fab and 500 μl of 0.95 mg ml$^{-1}$ active TMPRSS2, corresponding to an approximate molar ratio of 12:7:12, respectively, and incubated for 5 min at room temperature. The assembled complex was isolated by SEC on a Superdex 200 Increase 10/300 GL column (Cytiva) equilibrated in 50 mM HEPES pH 7.5 and 150 mM NaCl, concentrated to 9 mg ml$^{-1}$ and stored on ice until cryo-EM grid preparation.

For cryo-EM of the H1H7 Fab–anti-kappa nanobody–TMPRSS2-S441A complex bound to the NL63-S$_2$′ peptide, the same procedure was followed, except that 600 μM HCoV-NL63-S$_2$′ peptide with a C-terminal PEG6-Lys-biotin (described in Supplementary Table 1) was included during complex assembly. Instead of purification by SEC, the assembled complex was concentrated to 50 μl and washed twice using an Amicon Ultra centrifugal filter unit (100-kDa MWCO) with 400 μl of 50 mM HEPES pH 7.5 and 150 mM NaCl to remove unbound peptide and protein. Rapid concentrator-based washing was used in place of SEC to preserve the peptide–TMPRSS2 interaction, which exhibits rapid binding kinetics. The final complex was concentrated to 10 mg ml$^{-1}$ and stored on ice until cryo-EM grid preparation.

The HCoV-NL63-S$_2$′/HKU1-RBD + TMPRSS2-S441A complex was assembled by combining 200 μg of HCoV-NL63-S$_2$′/HKU1-RBD with 400 μg of TMPRSS2-S441A for 5 min at room temperature, followed by buffer exchange with 400 μl of 25 mM Tris and 150 mM NaCl and concentration using an Amicon Ultra centrifugal filter unit (100-kDa MWCO).

For cryo-EM grid preparation, 400 μl of 1.25 mg ml$^{-1}$ E-FIC$_S$ was incubated with 1.25 mg ml$^{-1}$ TMPRSS2-S441A for 5 min at room temperature, followed by removal of unbound TMPRSS2 by concentration to 50 μl, and washed five times using an Amicon Ultra centrifugal filter unit (100-kDa MWCO) with 400 μl of 50 mM Tris-HCl pH 7.5 and 150 mM NaCl. No bound TMPRSS2 was detected in the data, consistent with the rapid binding kinetics and low binding affinity of TMPRSS2 for the SARS-CoV-2 S$_2$′ peptide.

## BLI binding assays

All BLI experiments were performed at 30 °C rotating at 1,000 rpm; data were baseline-subtracted and the plots were fitted using Sartorius analysis software and plotted in GraphPad Prism 10.

Biotinylated peptides were loaded to a 0.5-nm shift onto SA biosensors prehydrated in 10× kinetics buffer and dipped in 1 μM TMPRSS2-S441A. For $K_D$ determination, biotinylated HCoV-NL63-S$_2$′ peptide or HCoV-NL63-S$_2$′/HKU1-RBD was immobilized to the surface of the biosensors and dipped in a 1:2 dilution series of TMPRSS2-S441A. Association and dissociation durations were each carried out for 300 s.

For the BLI assay using VN01H1, C13A7, C13B8, C13C9, C28F8, C77G12 and VP12E7 IgGs, each mAb was loaded to a 1-nm shift on anti-human Fc capture biosensors (Sartorious) that were prehydrated in 10× kinetics buffer for at least 10 min. The loaded tips were then baselined in 10× kinetics buffer and dipped into either 0.1 mg ml$^{-1}$ E-FIC$_{S-v1/3}$ or 0.05 mg ml$^{-1}$ SARS-CoV-2 HexaPro S to associate for 500 s and then dipped into 10× kinetics buffer to dissociate for 500 s.

For the BLI assays using the HR2 peptide, 1 μM of the biotinylated SARS-CoV-2 HR2 peptide (PDVDLGDISGINASVVNIQKEIDRLNEVAKN-LNESLIDLQEL-PEG6){Lys(biotin)}) was loaded to a 0.5-nm shift on streptavidin (SA) biosensors (Sartorius) that were prehydrated in 10× kinetics buffer for 20 min. Separately, E-FIC$_{S-v3}$ and TMPRSS2 were prepared by mixing E-FIC$_{S-v3}$ to a final concentration of 0.1 mg ml$^{-1}$ with either active (S441) TMPRSS2 or inactive (A441) TMPRSS2 to a final concentration of 200 nM and incubated for 30 min at room temperature. The HR2 peptide-loaded tips were baselined and then dipped into E-FIC$_{S-v3}$ preincubated or not with TMPRSS2 S441, TMPRSS2 A441 or 10× kinetics buffer as a baseline for 500 s (association phase).

The biosensors were dipped into 10× kinetics buffer for 500 s for the dissociation phase.

## Negative-stain EM and western blot sample preparation

SARS-CoV-2 $S_{ecto}$ R815 and R815H and the monomeric human ACE2 (peptidase) ectodomain were combined at a 1:3 molar ratio with a final S concentration of 666 nM and incubated on ice for 5 min before addition of active or inactive TMPRSS2 to a final concentration of 200 nM and final SARS-CoV-2 $S_{ecto}$ concentration of 656 nM before incubation on ice for 45 min. Samples were diluted to a final SARS-CoV-2 $S_{ecto}$ concentration of 20 μg ml$^{-1}$ and grids were stained with 2% uranyl formate. A fraction of the sample was also stopped with reducing loading dye containing 200 mM Tris pH 6.8, 8% SDS, 20% β-mercaptoethanol, 40% glycerol and 0.2% bromophenol blue and analyzed by western blot as described below. The remaining sample was digested with proteinase K at a final concentration of 10 μg ml$^{-1}$ and a final SARS-CoV-2 $S_{ecto}$ concentration of 328 nM for 30 min at 4 °C before stopping the reaction with the same reducing loading dye described above. These experiments were carried out in duplicate with two batches of proteins produced independently.

## Negative-stain EM data acquisition and processing

Grids were imaged on a Technai T12 transmission EM instrument at ×67,000 magnification to obtain representative micrographs of each grid. A total of 116, 115, 121, 115, 111 and 112 micrographs were collected for S, S + TMPRSS2-S441A, S + TMPRSS2, S + ACE2, S + ACE2 + TMPRSS2-S441A and S + ACE2 + TMPRSS2, respectively, on a Talos L120C transmission EM instrument at ×57,000 magnification using EPU data collection software (Thermo Fisher Scientific). Micrographs were imported into cryoSPARC[115] and the contrast transfer function (CTF) was determined by patch CTF estimation. Blob particle picking followed by two-dimensional (2D) classification was used to generate templates for repicking each dataset. Particles were subsequently extracted in boxes of 192 × 192 pixels and initially 2D classified into 100 classes with 60 initial iterations and two final iterations. Then, selected classes were 2D classified into 50 classes with the same parameters as the initial 2D classification and sorted into prefusion S, prefusion S with one ACE2 receptor bound, prefusion S with two ACE2 receptors bound, E-FIC and postfusion S. Each subset of particles was then 2D classified again into 10–25 classes with the same parameters, except for S, ACE2 and TMPRSS2-S441A, which had 60 final iterations for the ACE2-bound prefusion particles. For datasets for which ACE2 was present, each conformation was 2D classified again into ten classes with 60 final iterations and conformations were sorted by select 2D. Particles in each 2D class corresponding to each conformation were combined and initial ab initio three-dimensional (3D) reconstructions were generated followed by homogeneous refinement with a maximum alignment resolution of 15 Å and applying $C_3$ symmetry except for prefusion S with one and two ACE2 receptors bound, which were refined with $C_1$ symmetry. Particle counts were determined after combining the conformations after the third or fourth round of 2D classification for datasets in the absence or presence of ACE2, respectively. Maps were exported and modeled in ChimeraX and charts were rendered in GraphPad Prism10.

## Cryo-EM sample preparation, data acquisition and processing

For cryo-EM grid preparation, the following protein mixtures were used at the indicated concentrations with detergent to minimize preferred particle orientation: 2 mg ml$^{-1}$ NL63-$S_2'$/HKU1-RBD + TMPRSS2-S441A in 0.01% (w/v) fluorinated octyl maltoside (FOM; Anatrace), 9 mg ml$^{-1}$ H1H7 Fab + anti-kappa nanobody + TMPRSS2 S441 in 0.02% (w/v) FOM, 10 mg ml$^{-1}$ H1H7 Fab + anti-kappa nanobody + TMPRSS2-S441A + HCoV-NL63-$S_2'$ peptide in 0.02% (w/v) FOM or 10 mg ml$^{-1}$ E-FIC$_S$ + TMPRSS2-S441A (the latter of which was washed away during preparation) in 0.02% (w/v) FOM. Aliquots (3 μl) of each sample were applied

to freshly glow-discharged 2.0/2.0 UltraFoil gold grids (200-mesh; Quantifoil) and plunge-frozen using a Vitrobot Mark IV (Thermo Fisher Scientific) with a blot force of −1 and a blot time of 6 s, at 100% humidity and 23 °C. For the VN01H1 Fab-bound E-FIC complex, E-FIC was first prepared by incubating SARS-CoV-2 $S_{ecto}$ and monomeric ACE2 at a molar ratio of 1:1.5 for 40 min on ice before addition of a 1.2× molar excess of VN01H1 Fab and incubation for 5 min on ice. For cryo-EM grid preparation, 3 μl of the VN01H1 Fab-bound E-FIC–ACE2 complex at 0.6 mg ml$^{-1}$ were applied onto freshly glow-discharged 2.0/2.0 UltraFoil gold grids before plunge-freezing using a Vitrobot Mark IV with a blot force of 0 and a blot time of 5 s, at 100% humidity and 22 °C.

Data were acquired using Leginon[116] for the TMPRSS2-S441A–H1H7 Fab–anti-kappa nanobody complex and SerialEM[117] for the NL63-$S_2'$/HKU1-RBD + TMPRSS2-S441A, TMPRSS2-S441A–H1H7 Fab–anti-kappa nanobody–NL63 peptide complex, E-FIC$_{S-v1}$ complex, E-FIC$_{S-v3}$–VN01H1 complex and E-FIC$_{S-v3}$–C77G12 complex using an FEI Titan Krios transmission EM instrument operated at 300 kV and equipped with a Gatan K3 Summit direct electron detector and Gatan Quantum energy filter, operated in zero-loss mode with a 20-eV slit width. For the TMPRSS2-S441A–H1H7 Fab–anti-kappa nanobody complex and NL63-$S_2'$/HKU1-RBD + TMPRSS2-S441A complex, videos were collected at a dose rate of 15 counts per pixel per second, with 75 frames per video and an exposure time of 40 ms per frame, corresponding to a physical pixel size of 0.83 Å. For the TMPRSS2-S441A–H1H7 Fab–anti-kappa nanobody–NL63 peptide complex, E-FIC$_{S-v1}$ complex, E-FIC$_{S-v3}$–VN01H1 complex and E-FIC$_{S-v3}$–C77G12 complex, videos were collected at a dose rate of 15 counts per pixel per second, with 99 frames per video and an exposure time of 50 ms per frame, corresponding to a physical pixel size of 0.83 Å. A defocus range of 0.5 to 2.8 μm was used. Each dataset was collected in a single session without stage tilt, whereas the NL63-$S_2'$/HKU1-RBD dataset was collected across two sessions, also without tilt. Video frame alignment and twofold pixel binning were performed using Warp[118]. CTF estimation (patch CTF), particle picking (Topaz[119]) and extraction were carried out in cryoSPARC[115]. Ab initio reconstruction and heterogeneous refinement with multiple classes were performed iteratively in cryoSPARC to generate high-quality initial models. Reference-free 2D classification was used to select well-defined particle images before nonuniform refinement in cryoSPARC[120]. For each dataset, Bayesian polishing was performed in RELION[121] using unbinned particle stacks, with data export assisted by pyem[122], followed by 2D classification to select well-defined polished particles. At this stage, particles from two sessions for the HCoV-NL63-$S_2'$/H KU1-RBD dataset were merged. For the TMPRSS2-S441A–H1H7 Fab–anti-kappa nanobody complex and TMPRSS2-S441A–H1H7 Fab–anti-kappa nanobody–HCoV-NL63-$S_2'$ peptide complex, nonuniform refinement with per-particle defocus refinement was then performed in cryoSPARC. For the HCoV-NL63-$S_2'$/HKU1-RBD dataset, an additional round of ab initio and heterogeneous refinement preceded final nonuniform refinement with per-particle defocus correction[120,123]. For the E-FIC$_{S-v1}$ dataset, initial refinements were overfitted, likely as a consequence of the narrow and elongated particle shape. To overcome overfitting, after a round of global nonuniform refinement, local refinement using a global map and mask to enable pose/shift Gaussian priors (penalties for deviations from the previous alignment) during alignment improved map resolution. This was subsequently followed by standalone local defocus refinement and a second round of local refinement using a global map and Gaussian priors. Subsequent filtering of low-quality particles, defined by particle scale, 3D alignment error (greater than 40,162 error value), difference in shift and difference in pose. Lastly, for the E-FIC$_{S-v1}$ dataset, a third round of local refinement using a global map and mask to enable pose/shift Gaussian priors. For the E-FIC$_{S-v3}$–VN01H1 complex and E-FIC$_{S-v3}$–C77G12 complex datasets, nonuniform refinement with per-particle defocus refinement was then performed in cryoSPARC. Focused masks separating each complex into two regions were generated in UCSF Chimera and used for particle subtraction followed by local refinement in cryoSPARC.

Data for the VN01H1 Fab-bound E-FIC–ACE2 complex were acquired using an FEI Titan Krios transmission EM instrument operated at 300 kV and equipped with a Gatan K3 direct detector and Gatan Quantum GIF energy filter, operated in zero-loss mode with a slit width of 20 eV. Automated data collection was carried out using Leginon[116] at a nominal magnification of ×105,000 with a pixel size of 0.835 Å. The dose rate was adjusted to 10.5 counts per pixel per second and each video was acquired in counting mode fractionated in 100 frames of 40 ms. A total 10,807 micrographs were collected with a defocus range between 0.2 and 3 µm and stage tilt angles of 0°, 15° and 25°. Video frame alignment was carried out using cryoSPARC[115] and RELION[121]. Then, cryoSPARC[115] was used for video frame alignment (patch motion correction), estimation of contrast transfer function (patch CTF), particle picking (blob picker), particle extraction and reference-free 2D classification to select well-defined particle images. These particles were used for ab initio reconstruction and heterogeneous refinement in cryoSPARC[115]. To further improve particle picking, the selected particles were used to carry out four rounds of Topaz[119] training and extraction combined with reference-free 2D classification, ab initio reconstruction and heterogeneous refinements in cryoSPARC[115]. At each round, selected particle sets were combined and duplicated particles were removed using a minimum distance cutoff of 50 Å. The selected particles were extracted with a box size of 384 × 384 pixels at 1.392 Å per pixel and 3D refinement was carried out using nonuniform refinement along with per-particle defocus refinement in cryoSPARC[120]. To run Bayesian polishing in RELION[121], we realigned video frames using RELION's own MotionCorr followed by subsequent reextraction in RELION of the particles used in cryoSPARC using pyem[122]. The reextracted particles were imported in cryosPARC for subsequent patch CTF estimation and nonuniform refinement with per-particle defocus refinement. After exporting this particle stack and angles/shifts to run Bayesian polishing, the particle stack was imported back into cryoSPARC to run nonuniform refinement with per-particle defocus refinement, yielding a reconstruction at 3.8-Å resolution comprising 389,200 particles. To further improve map resolvability, we used local refinement in cryoSPARC[115] with two soft masks encompassing the membrane-proximal region of E-FIC and the VN01H1 Fab variable domains or the membrane-distal region of E-FIC yielding reconstructions at 3.3-Å resolution. Resolution estimates used the gold-standard Fourier shell correlation (FSC) criterion of 0.143 and FSC curves were corrected for the effects of masking using high-resolution noise substitution[124,125].

### Model building and refinement
Initial models used for model building were structure predicted by AlphaFold3 (ref. [110]) and rigid-body docked into the maps using UCSF Chimera[126]. For the TMPRSS2-S441A–H1H7 Fab–anti-kappa nanobody–NL63-S2′ peptide complex, we used the TMPRSS2-S441A–H1H7 Fab–anti-kappa nanobody structure as a starting model. Unmodified unsharpened and sharpened maps were used during model building in Coot[127,128] and refinement. Further model optimization was performed using ISOLDE[129] (implemented in UCSF ChimeraX[130]) with an AMBER force field to enable molecular-dynamics-guided refinement. Rosetta[131,132] was subsequently used to further improve geometry and fit to density.

For the VN01H1 Fab-bound E-FIC structure, the VN01H1 Fab (PDB 7SKZ) and E-FIC (PDB 8Z7P) structures were rigid-body docked into the unsharpened map using UCSF Chimera[126]. Coot[127,128] was used to rebuild atomic models into the sharpened and unsharpened maps obtained from local refinements of the membrane-proximal and membrane-distal regions and refinement was carried out using PHENIX[133] with strict $C_3$ symmetry. Final models were validated using MolProbity[134] and PHENIX[133].

Model building in Coot[127,128] was assisted by using maps enhanced with ARdecon[135] and EMready[136]. Unmodified unsharpened and sharpened maps were deposited. Final models were validated using MolProbity[134], EMringer[137], Privateer[138] and PHENIX[133]. Structural figures were generated using UCSF ChimeraX[130]

### Peptidase assay
TMPRSS2 (0.00185 mg ml⁻¹) was preincubated at 22 °C for 2 min with a threefold dilution series of either H1H7 mAb (starting at 0.2 mg ml⁻¹) or camostat (starting at 250 nM). Reactions were initiated by addition of 0.1 mM fluorescent substrate (Dabcyl-RRARSVASQSI-(Glu)EDANS; GenScript) in black, half-area 96-well plates (Greiner Bio-One, Fluotrac) with a final volume of 100 µl per well. Fluorescence was monitored every 2 min for 30 min at 22 °C using an Agilent BioTek Neo2 microplate reader (excitation, 336 nm; emission, 490 nm). Initial reaction velocities were calculated from the linear slope over the first 10 min, corresponding to <5% substrate conversion, and normalized to the uninhibited TMPRSS2 control. Velocities represent the mean ± s.d. of two technical replicates. Data were plotted and fitted using GraphPad Prism (version 10.1.1).

### SDS–PAGE
TMPRSS2-mediated cleavage of SARS-CoV-2 Secto was observed at 1 µM, 500 nM, 250 nM, 125 nM, 62.5 nM, 31.3 nM, 16.1 nM, 8.0 nM and 4.0 nM TMPRSS2. SARS-CoV-2 Secto and monomeric hACE2 were combined at a 1:3 molar ratio, with a final SARS-CoV-2 Secto concentration of 833 nM in 50 mM Tris pH 7.4 and 150 mM NaCl to a final volume of 15 µl, and incubated for 5 min at room temperature. Active TMPRSS2 was added to the designated concentrations to a final SARS-CoV-2 Secto concentration of 625 nM and volume of 20 µl and samples were incubated at room temperature for 15 min. Reactions were stopped with 6.6 µl of reducing loading dye containing 200 mM Tris pH 6.8, 8% SDS, 20% β-mercaptoethanol, 40% glycerol and 0.2% bromophenol blue. Next, 26.6 µl of samples were run on 4–20% precast polyacrylamide gel (Bio-Rad) with 4 µl of Precision Plus Kaleidoscope prestained protein ladder (Bio-Rad) at 80 A for 18 min for one gel. Gels were stained with InstantBlue Coomassie protein stain (Abcam) and gently agitated at room temperature. Gels were destained with MilliQ water overnight, agitating at room temperature. The presumed S2′ cleavage product appeared at around 100–200 nM TMPRSS2 with minimal nonspecific cleavage.

TMPRSS2-mediated cleavage of SARS-CoV-2 $S_{ecto}$ cleavage was observed at 4 °C, 25 °C and 37 °C at various time points. SARS-CoV-2 $S_{ecto}$ and monomeric hACE2 were combined at a 1:3 molar ratio, with a SARS-CoV-2 $S_{ecto}$ concentration of 1.17 µM in 50 mM Tris pH 7.4 and 150 mM NaCl, and incubated for 5 min at 4 °C, 25 °C and 37 °C, before the addition of TMPRSS2 and 10 µl of sample to 4 µl of SDS–PAGE loading dye containing 200 mM Tris pH 6.8, 8% SDS, 20% β-mercaptoethanol, 40% glycerol and 0.2% bromophenol blue, designated as the 0-min time point. Active TMPRSS2 was added to a final concentration of 100 nM and both samples were incubated at designated temperature, with a final SARS-CoV-2 $S_{ecto}$ concentration of 1.16 µM. Then, 10 µl of reaction was added to 4 µl loading dye at 3, 6,12, 24, 40, 60 and 90 min to stop the reaction. SARS-CoV-2 $S_{ecto}$ at a concentration of 1.17 µM and TMPRSS2 at a concentration of 100 nM were combined and incubated for 90 min at 4 °C, 25 °C and 37 °C and 10 µl of sample was added to loading dye to show that TMPRSS2 had no observed cleavage $S_2′$ cleavage in the absence of ACE2. Monomeric hACE2 at a concentration of 3.5 µM and TMPRSS2 at a concentration of 100 nM were combined and incubated for 90 min at 4 °C, 25 °C and 37 °C and 10 µl of sample was added to loading dye to show that TMPRSS2 had no observed cleavage of the monomeric hACE2. Next, 13 µl of samples were run on 4–20% precast polyacrylamide gel (Bio-Rad) with 4 µl of Precision Plus Kaleidoscope prestained protein ladder (Bio-Rad) at 80 A for 18 min for one gel and 35 min for two gels. Gels were stained with InstantBlue Coomassie protein stain (Abcam), agitating at room temperature. Gels were destained with MilliQ water overnight, agitating at room temperature. Gels were imaged on a transilluminator.

## Western blot

Samples were prepared as described in the negative-stain EM and SDS–PAGE preparation sections. Gels were electroblotted onto Trans-Blot Turbo Mini 0.2-μm PVDF transfer packs (Bio-Rad) at 1.3 A for one blot and 2.5 A for two blots for 7 min. Blots were blocked with 5% milk for 2 h and washed three times with TBST containing 20 mM Tris pH 7.5, 150 mM NaCl and 0.1% v/v Tween-20, agitating for 5 min. Primary antibodies were diluted in TBST, added to the blots and incubated overnight at 4 °C with agitation. These included MAB10540 (R&D systems) used at a final concentration of 2 μg ml$^{-1}$ to detect S$_1$, humanized B6 (ref. [66]) IgG and 76E1 (ref. [20]) IgG used at a final concentration of 10 μg ml$^{-1}$ to detect S$_2$ and SARS-CoV-2 S$_2$ polyclonal rabbit antibody (SinoBiological) at a 1:1.000 dilution used to detect the proteinase-K-resistant six-helix bundle. Blots were washed and agitated three times for 5 min with TBST before adding the secondary antibodies (donkey-anti-mouse-680 (Licor) for MAB10540, goat anti-human-680 (Licor) for humanized B6 (ref. [66]) IgG or 76E1 (ref. [20]) IgG and goat anti-rabbit-680 (Licor) for polyclonal SARS-CoV-2 S$_2$ antibody) and agitated for 2 h at room temperature. Three additional washes with TBST were carried out for 5 min with shaking and images were acquired on a Licor 700-channel for 2 min.

## SARS-CoV-2 pseudotyped virus production and neutralization of entry into Vero E6 and Vero E6-TMPRSS2 cells

Lenti-X 293T cells were seeded in 10-cm dishes for 80% next-day confluency. The next day, cells were transfected with a plasmid encoding the SARS-CoV-2 S protein (isolate Wuhan-Hu-1, C-terminal truncation of 19 aa) using the transfection reagent TransIT-Lenti according to the manufacturer's instructions. Then, 1 day after transfection, cells were infected with the replication-deficient VSV-luc(VSV-G) (Kerafast), in which the VSV-G open reading frame (ORF) was replaced by a luciferase ORF, at a multiplicity of infection (MOI) of 3–10. The cell supernatant containing SARS-CoV-2 pseudotyped virus was collected on day 2 after transfection, centrifuged to remove cellular debris, aliquoted and frozen at −80 °C.

Infectious titer of the SARS-CoV-2 pseudotyped virus preparation was quantified using Vero E6 cells seeded at 20,000 cells per well in clear-bottom black-walled 96-well plates the previous day. Cells were infected with serially diluted pseudotyped virus in 50 μl of medium for 1 h at 37 °C. An additional 100 μl of medium was added and cells were incubated overnight at 37 °C. Cells were fixed with 4% PFA, permeabilized with Triton X-100 and stained with an anti-VSV-N primary antibody (Absolute Antibody), a goat anti-mouse AlexaFluor647 or AlexaFluor488 secondary antibody (Thermo Fisher Scientific) and Hoechst dye (Thermo Fisher Scientific) in blocking buffer. The number of VSV N+ cells was quantified using a Cytation5 plate reader to calculate pseudotyped virus titers.

For neutralization of SARS-CoV-2 S VSV pseudotyped viruses, Vero E6 or Vero E6-TMPRSS2 cells were seeded into clear-bottom black-walled 96-well plates at 20,000 cells per well and cultured overnight at 37 °C. The next day, nine-point fourfold serial dilutions of mAbs were prepared in medium. The medium was removed from cells and 50 μl of mAb dilution was added to the cells for a 2-h preincubation. SARS-CoV-2 S pseudotyped VSVs were diluted at an MOI of 0.1 in medium and added 1:1 to fresh mAb dilutions. After preincubation, the medium was removed from the cells and 50 μl of virus–mAb mixtures were added. Then, 1 h after infection, 100 μl of medium was added to all wells. After 20–24 h of incubation at 37 °C, the medium was removed and 50 μl of Bio-Glo reagent (diluted 1:1 in Dulbecco's PBS) was added to each well. The plates were incubated at room temperature for 15 min and luminescence was read on an Ensight plate reader (Perkin-Elmer). To control for background, the mean of the relative luminescence unit (RLU) values in uninfected wells was calculated and subtracted from all data points. The percentage inhibition was calculated and normalized to the mean of the RLUs in virus-only wells (no antibody). Data were

analyzed and graphed using GraphPad Prism software (version 9.0 or later). The half-maximal effective concentration (EC$_{50}$) values were calculated using a nonlinear regression model (variable slope model, four parameters) of log(inhibitor) versus response and the EC$_{50}$ values were interpolated from the curve at $y = 50$.

## SARS-CoV-2 pseudotyped virus production and neutralization of entry into Calu-3 cells

MERS-CoV, SARS-CoV-1 and SARS-CoV-2 Wu/G614 S VSV pseudoviruses were produced using HEK293T cells seeded on BioCoat cell culture dish coated with 100 mm of poly(D-lysine) (Corning). Membrane-anchored full-length S glycoprotein constructs were used to pseudotype VSV as previously described. In brief, SARS-CoV-2 Wu/G614 S (NC_045512) has native signal peptide, followed by residues 1–1252 with the D614G substitution and 21 residues deleted from the C terminus[139] (BEI, NR-53742). SARS-CoV-1 S (AY278741) has a CD5 signal peptide, followed by residues 12–1255 with a C-terminal C9 tag (TETSQVAPA)[140]. MERS-CoV S (AFS88936.1) has a native signal peptide, followed by residues 1–1353 with a C-terminal C9 tag (TETSQVAPA)[17]. Cells were transfected with respective S constructs using Lipofectamine 2000 (Life Technologies) in Opti-MEM transfection medium. After 5 h of incubation at 37 °C with 5% CO$_2$, cells were supplemented with DMEM containing 10% FBS. The next day, cells were infected with VSV (G*ΔG-luciferase) for 2 h, followed by five washes with DMEM before the addition of anti-VSV-G antibody (I1-mouse hybridoma supernatant; ATCC, CRL-2700) and medium. After 18–24 h of incubation at 37 °C with 5% CO$_2$, pseudoviruses were collected and cell debris was removed by centrifugation at 3,000$g$ for 10 min. Pseudoviruses were further filtered using a 0.45-μm syringe filter (GenClone) and concentrated tenfold for a neutralization assay before storage at −80 °C.

Then, 2 days before the pseudovirus neutralization assay, Calu-3 cells in DMEM/F-12 GlutaMAX supplement (Gibco) were plated in 96-well plates with a cell density of 40,000 cells per well. Cells were incubated at 37 °C with 5% CO$_2$ for 2 days. On the day of neutralization assay, in a half-area 96-well plate (Greiner), threefold serial dilutions of H1H7 IgG (starting concentration of 50 μg ml$^{-1}$, 44 μl in total volume per well) were prepared. The medium was removed from cells and 40 μl of the mixture was transferred to the cell, followed by a 1-h incubation at 37 °C with 5% CO$_2$. In another half-area 96-well plate, threefold serial dilutions of H1H7 IgG (22 μl in total volume per well) were mixed with an equal volume of EMEM (ATCC, 30-2003) with diluted pseudoviruses (final concentration of 50 μg ml$^{-1}$ H1H7 IgG after mixing). Pseudovirus dilutions used were predetermined for each pseudovirus batch on the basis of entry levels (RLUs). After 1 h of preincubation of cells with H1H7 IgG, the medium was removed and replaced with 40 μl of mixture containing H1H7 IgG and pseudovirus. Wells with cells only and cells with pseudovirus added were prepared to serve as 0% infectivity and 100% infectivity, respectively. Cells were incubated for 90 min at 37 °C with 5% CO$_2$, followed by the addition of 40 μl of EMEM supplemented with 20% FBS. After 22–25 h of incubation at 37 °C with 5% CO$_2$, the medium was removed from the cells and 80 μl of One-Glo-EX substrate (Promega) was added to each well, before incubating on a plate shaker in the dark for 10–15 min and reading the luciferase signal using a BioTek Neo2 plate reader. RLUs were plotted and normalized in Prism (GraphPad). Prism nonlinear regression with '[inhibitor] versus response with variable slope (four parameters)' was used to fit the curve with the bottom constraint and the top constraint set to >0 and <100, respectively. Each neutralization was performed with 3–4 technical replicates and three biological replicates per pseudovirus.

## Authentic SARS-CoV-2 virus infection using Vero E6 and Vero E6-TMPRSS2 cells

Authentic SARS-CoV-2 viral isolates were obtained from BEI Resources (WA-1, NR-52281; BA.2, NRS-56511; XBB.1.5, NR-59104; CH.1.1, NRS-59204). To propagate SARS-CoV-2 viral stocks, Vero E6-TMPRSS2 cells

were seeded at $1 \times 10^7$ cells in T175 flasks in growth medium and infected the next day at an MOI of 0.01 in infection medium (DMEM and 2% BSA). Virus was adsorbed for 1 h at 37 °C. Virus inoculum was removed, flasks were washed twice with PBS, 20 ml of infection medium was added to the cells and flasks were incubated at 37 °C. Supernatants were collected at 72 h after infection and centrifuged at 500g for 5 min, followed by a second centrifugation at 2,000g for 10 min. Clarified supernatants were then aliquoted and stored at −80 °C. Virus titers were determined using a plaque assay on Vero E6-TMPRSS2 cells, using standard methods. Briefly, tenfold dilutions of virus stock were incubated in six-well plates with 2.4% colloidal cellulose overlay for 24–30 h. Cells were fixed with 4% PFA for 30 min at RT, permeabilized with 0.25% Triton X-100 and stained with anti-SARS-CoV-2 nucleocapsid antibody at 1:2,000 in 2% BSA in PBS and goat anti-rabbit IgG HRP secondary antibody at 1:4,000 in 2% BSA in PBS. Plaque-forming units (PFU) were visualized with TrueBlue reagent. All viral sequences were verified by deep sequencing of the viral stocks.

For infection and neutralization assays, Vero E6 or Vero E6-TMPRSS2 cells were seeded into clear-bottom black-walled 96-well plates at 20,000 cells per well and cultured overnight at 37 °C. The next day, the medium was removed and cells in the treatment group were pretreated with 5 µg ml⁻¹ H1H7 mAb in 50 µl of growth medium for 2 h at 37 °C. SARS-CoV-2 authentic virus stock was diluted in infection medium (DMEM + 2% BSA) for a final concentration of 200 PFU per well (MOI = 0.01). H1H7 mAbs at a 5 µg ml⁻¹ final concentration were added to the virus mix for the treatment group. The medium was removed from the cells, virus mixtures were added and cells were incubated at 37 °C. At 16–28 h after infection, cells were fixed with 4% PFA for 30 min at room temperature and then washed three times with PBS to remove residual PFA. The cells were permeabilized with 100 µl of 0.25% Triton X-100 in PBS for 30 min at room temperature, followed by three washes with PBS. Cells were incubated with 50 µl of anti-SARS-CoV-2 nucleocapsid antibody (Sino Biologicals, 40143-R001) at 1:2,000 for 1 h at room temperature. Plates were washed three times with PBS and then incubated for 1 h at room temperature with 50 µl per well of goat anti-rabbit IgG Alexa647 (Invitrogen, A-21245) secondary antibody at a final dilution of 1:1,000 mixed with 2 µg ml⁻¹ Hoechst dye. After washing three times with PBS, 200 µl of fresh PBS was added for imaging. Plates were imaged on a Cytation5 plate reader. Whole-well images were acquired (12 images at ×4 magnification per well) and nucleocapsid-positive cells were counted using the manufacturer's software.

## High-throughput SARS-CoV-2 virus inhibition assay

A high-content fluorescence microscopy approach was used to assess the ability of H1H7 IgG to inhibit SARS-CoV-2 infection in cells that stably express human ACE2, TMPRSS2 and SLC6A19 (TASL-19)[87]. The antibody was initially diluted in cell culture medium (MEM and 2% FCS) to make 4× working stock solutions and then serially diluted in the above media to achieve a threefold dilution series. The diluted antibody was mixed in a 1:1 ratio with nuclear-stained (Invitrogen, R37605; final concentration at 2.5% (v/v)) TASL-19 cells and transferred to a 384-well plate (Corning, CLS3985) such that each well contained 10,000 cells and 2× working antibody solution in a volume of 30 µl of MEM and 2% FCS per well. Each condition or dilution was run in four technical replicates for each virus. Nafamostat and human eCD4-Ig (courtesy of M. Farzan[141]) were used as positive and negative controls for TMPRSS2 inhibitory activity, respectively. The plates containing the cells and the antibodies or nafamostat were incubated for 30 min at 37 °C in 5% CO₂, after which 30 µl of virus solution at 5× median virus effective dose[142] was added to the wells. In brief, virus effective dose is a machine-based 50% tissue culture infectious dose (TCID₅₀) titer measurement that uses the dose-dependent loss of nuclei through viral replication to plot sigmoidal curves to then interpolate the midpoint dilution (50% loss of nuclei). Plates were incubated at 37 °C in 5% CO₂ for a further 48 h, after which cell nuclei were enumerated using an INCell

Analyzer 2500HS high-content microscope. The percentage inhibition was calculated with the formula %$N = (D − (1 − Q)) \times 100/D$, as previously described[143]. $Q$ is a well's nucleus count divided by the average count for uninfected controls (defined as having 0% inhibition) and $D = 1 − Q$ for the average count of positive infection controls (defined as having 0% inhibition). Nonlinear curve fit was performed using [inhibitor] versus normalized response (variable slopes) in GraphPad Prism. All viruses were obtained using remnant diagnostic swabs under University of New South Wales (NSW) ethics approval iREC5100. All viruses were expanded using TASL-19 cells by infecting $5 \times 10^6$ cells with a MOI of 0.025 and then isolated by collecting the supernatant after centrifugation at 1,500g for 10 min at 4 °C and then storage at −80 °C. All expanded isolates were in culture less than 1 week following the initial culture of primary diagnostic swabs. As part of this process, all isolates are sequenced with publicly available GISAID codes for each isolate used. Viral isolation was in collaboration with NSW Health Pathology as part of ongoing genotype-to-phenotype surveillance, where diagnostics and whole-genome sequencing initially identified clinical material for isolate expansion.

## Mice used for SARS-CoV-2 challenge

Virus inoculations were performed under anesthesia that was induced and maintained with ketamine hydrochloride and xylazine and all efforts were made to minimize animal suffering. hTMPRSS2-KI (C57 BL/6J-*Tmprss2*em1(TMPRSS2)Synbl/J; strain 036900) and TMPRSS2-KO mice (B6.129-*Tmprss2*tm1Psn/J; strain 026196) were originally obtained from Jackson Laboratories and bred in-house for SARS-CoV-2 challenge. These animals were housed in a pathogen-free animal facility at Washington University in St. Louis and used with an age range of 5–8 weeks. Animals were housed in HEPA-filtered microisolator caging units that contained up to five animals per cage. Environmental conditions for the animal room were set to maintain temperature between 68 °F to 74 °F, a relative humidity of 30–60% and a 12-h light–dark cycle. Six female and two male mice were used for the isotype control group for analysis at 4 and 6 dpi, respectively. For the hTMPRSS2-KI mice, we used eight female mice (4 dpi) and one female and seven male mice (6 dpi) for the H1H7 25 mg kg⁻¹ group, four female and four male mice (4 dpi) and eight female mice (6 dpi) for the H1H7 10 mg kg⁻¹ group and six female and two male mice (4 dpi) and six female and two male mice (6⁻¹dpi) for the H1H7 5 mg kg⁻¹ group.

The SARS-CoV-2 B.1.351 strain was isolated from an infected individual. Viruses were propagated on Vero E6-TMPRSS2 cells and subjected to deep sequencing to confirm the presence of expected substitutions. Viral titer was determined by focus-forming assay (FFA) as described[144]. Viral infections were performed through intranasal inoculation with the quantity of virus indicated in each figure legend.

## Measurement of SARS-CoV-2 viral RNA

On the indicated dpi, mice were killed and organs were collected. Tissues were weighed and homogenized with zirconia beads in a MagNA Lyser instrument (Roche Life Science) in 1 ml of DMEM supplemented with 2% heat-inactivated FBS. Tissue homogenates were clarified by centrifugation at approximately 10,000g for 5 min and stored at −80 °C. RNA was extracted using the MagMax mirVana total RNA isolation kit (Thermo Fisher Scientific) on the Kingfisher Flex extraction robot (Thermo Fisher Scientific). RNA was reverse-transcribed and amplified using the TaqMan RNA-to-CT one-step kit (Thermo Fisher Scientific). Reverse transcription was carried out at 48 °C for 15 min followed by 2 min at 95 °C. Amplification was accomplished over 50 cycles as follows: 95 °C for 15 s and 60 °C for 1 min. Copies of the SARS-CoV-2 *N* gene RNA in samples were determined using a previously published assay[144]. Briefly, a TaqMan assay was designed to target a highly conserved region of the *N* gene SARS-CoV-2 strains: forward primer, ATGCTGCAATCGTGCTACAA; reverse primer, GACTGCCGCCTCTGCTC; probe, /56-FAM/TCAAGGAAC/ZEN/AACATTGCCAA/3IABkFQ/. The

corresponding *N* region was included in an RNA standard to allow for copy-number determination down to ten copies per reaction. The reaction mixture contained final concentrations of primers and probe of 500 and 100 nM, respectively.

## SARS-CoV-2 plaque assays

Vero E6-TMPRSS2-hACE2 cells were seeded at a density of $1 \times 10^5$ cells per well in 24-well tissue culture plates. The following day, medium was removed, replaced with 200 µl of homogenate to be titrated and diluted serially in DMEM supplemented with 2% FBS. Then, 1 h later, 1 ml of methylcellulose overlay was added. Plates were incubated for 72 h and then fixed with 4% paraformaldehyde (final concentration) in PBS for 20 min. Plates were stained with 0.05% (w/v) crystal violet in 20% methanol and washed twice with distilled, deionized water before plaque enumeration.

## SARS-CoV-1 challenge

hTMPRSS2-KI mice (strain C57BL/6Hsd-*Tmprss2*$^{em1(TMPRSS2)Env}$) were obtained from Inotiv (formerly Envigo). Each group comprised eight animals (half male and half female) aged 5–8 weeks. Following transfer to ABSL-3 for dosing and subsequent challenge with SARS-CoV-MA15 (ref. [102]), the animals were housed in HEPA-filtered microisolator caging units up to four animals per cage. Environmental controls for the animal room were set to maintain temperature between 68 °F and 79 °F, a relative humidity of 30–70% and a 12-h light–dark cycle. SARS-CoV-MA15 virus challenge was administered intranasally on anesthetized animals (80 mg kg$^{-1}$ ketamine and 5 mg kg$^{-1}$ xylazine) and performed in a type II biosafety cabinet in an ABSL-3 suite. The mice administered a challenge dose of $2.5 \times 10^4$ TCID$_{50}$.

## Quantitation of viral load in lungs and nasal turbinates using TCID$_{50}$ assay

The infectious viral titers in lung samples and nasal turbinates were determined using a TCID$_{50}$ assay. At necropsy on days 2 and 5 after infection, lungs and nasal turbinates were collected and snap-frozen. Before testing the tissues, samples were thawed and then homogenized on ice in 500 µl of medium (DMEM + 2% FBS and Puro + P/S) for approximately 20 s using a hand-held tissue homogenizer. The samples were centrifuged 2,000*g* at 4 °C for 10 min) to remove cellular debris, pipetted through a 100-µm strainer back into the vial and stored on wet ice for same-day testing.

Briefly, clear, flat-bottom 96-well culture microplates were seeded with Vero E6 cells at $2.5 \times 10^4$ cells per well in growth medium (DMEM + 10% FBS + gentamicin) per well. Then, 20 µl of processed tissue sample (or positive or negative control) was added to the top row of the plate in quadruplicate (initial 1:10 dilution), mixed by pipetting and then serially diluted down the rows by 20 µl (tenfold dilution), changing pipette tips between each sample. Plates were incubated at 37 °C in 5% $CO_2$ for 4 days. After incubation, the presence or absence of cytopathic effects was noted. The TCID$_{50}$ value was calculated using the Read–Muench formula.

## MS analysis of TMPRSS2-mediated cleavage of SARS-CoV-2 S

To generate samples for MS analysis, 666 nM SARS-CoV-2 S$_{ecto}$ was incubated on ice for 5 min with or without 2,000 nM human ACE2 in a total volume of 498 µl containing 50 mM Tris-HCl pH 7.5 and 150 mM NaCl (yielding the E-FIC conformation). Each reaction was subsequently divided into three 160-µl aliquots: one supplemented with 4 µl of buffer, one supplemented with 4 µl of 8,200 nM TMPRSS2-S441A (inactive) and one supplemented with 4 µl of 8,200 nM TMPRSS2 S441 (active). The resulting mixtures were incubated on ice for 45 min. At this stage, 5 µl from each reaction was removed and flash-frozen in liquid nitrogen for later confirmation of conformation by negative-stain EM. The samples in the remaining reaction volumes were reduced by the addition of 3.3 µl of 50 mM TCEP and heated to 95 °C for 5 min. After

cooling to room temperature, 5 µl of PNGase F (New England Biolabs, P0704S) was added and the reactions were incubated at 37 °C for 20 h to achieve deglycosylation. Following incubation, all samples were flash-frozen in liquid nitrogen and stored at −80 °C until analysis.

All samples were mixed with a final concentration of 1% (w/v) sodium deoxycholate (SDC) and 100 mM HEPES buffer pH 8.5, where the disulfide bonds were reduced and cysteines were carbamidomethylated using a final concentration of 10 mM TCEP and 40 mM chloroacetamide. An overnight digestion at 37 °C using GluC (Sequencing grade, Promega) was performed with a final ratio of 1:50 (w/w) protease to protein. The resultant peptides were desalted using Strata-X cartridges (Phenomenex); the desalting cartridge was conditioned with 1 ml of acetonitrile followed by 1 mL 0.1% formic acid (FA) in water. Peptides were acidified with formic acid causing the SDC to precipitate and the supernatant was loaded, followed by a 1-ml wash with 0.1% FA in water. Peptides were eluted with 400 µl of 0.1% FA in 80% acetonitrile and dried by vacuum centrifugation. Samples were resuspended in 0.1% FA in water at a 1 µg µl$^{-1}$ concentration before analysis.

The peptides were analyzed on an Orbitrap Ascend Tribrid MS instrument (Thermo Fisher Scientific), coupled to a Vanquish Neo UHPLC (Thermo Fisher Scientific). First, samples were loaded on a C18 trap column (300 × 5 mm, 5-µm particles, PepMap Neo) using a flow rate of 300 nl min$^{-1}$ with buffer A (0.1% FA). Subsequently, samples were linearly eluted using an 85-min gradient ranging from 2.2% buffer B (99.9% acetonitrile with 0.1% FA) to 28% buffer B on a C18 Aurora Ultimate UHPLC analytical column (75 µm × 25 cm, 1.7-µm particles, IonOpticks). The column was washed by alternating between 99% and 2% B over 5 min to bring the total run time to 90 min inclusive. Ionization was performed using a Nanospray Flex source held at +2.0 kV compared to the ground and the ion transfer tube was set to 275 °C. MS1 scans were collected in the Orbitrap (*m/z* 375–2,000) with a normalized AGC target of 200% ($8 \times 10^5$ charges), a maximum injection time of 59 ms, and a resolution of 30,000 at 200 *m/z*. Monoisotopic precursor selection was enabled with precursor charge states between 2 and 5 selected for data-dependent MS/MS scans and dynamic exclusion applied for 30 s after two counts. Precursor ions were isolated with a 0.7 *m/z* quadrupole filter and fragmented by higher-energy collisional dissociation at a normalized collision energy of 30% with a normalized automatic gain control target of 200% ($1 \times 10^5$ charges), maximum injection time of 59 ms and resolution of 30,000 at *m/z* 200.

Raw data were searched using the 'basic-search' workflow in Frag-Pipe (version 23.1)[145,146]. A FASTA file was created, containing the four proteins that could be present in the combination of samples: S$_{ecto}$, ACE2, TMPRSS2 S441 and TMPRSS2 A441. Decoy sequences (reversed) and common contaminants were appended to the protein list by Frag-Pipe before searching. A GluC search (C-terminal cleavage after aspartic acid and glutamic acid residues but not after proline) was performed using MSFragger with semispecific (N terminus or C terminus) settings with two missed cleavages allowed. A fixed modification was set as carbamidomethyl on C (+57.02146 Da) and variable modifications were set as oxidation on M (+15.9949 Da, three maximum occurrences) and protein N-terminal acetylation (+42.0106, one maximum occurrence). Upon review of the peptides detected with the semispecific search, the most commonly cleaved residues other than cleavage at aspartic acid and glutamic acid were identified as asparagine, arginine and lysine. A secondary search was performed by including cleavage at these sites with two missed cleavages, as well as cleavage by GluC with two missed cleavages. The other parameters were not changed between the semispecific and secondary search. Peptides from the tailored secondary search were verified using Skyline (version 25.1)[147] and only those that had sufficient sequence coverage and high-quality MS1 chromatograms were included. Sequence alignments were performed with DigDig[148], spectral annotation was performed with the Interactive Peptide Spectral Annotator[149] and all other data processing was performed in R (version 4.5.1).

## Statistics and reproducibility

All statistical tests were performed as described in the indicated figure legends using GraphPad Prism. The number of independent experiments performed is indicated in the relevant figure legends and the Methods. All replicates were reproducible and no data were excluded from the analysis. No sample size calculations were performed to power each study. Instead, sample sizes were determined on the basis of prior in vivo virus challenge experiments. Randomization and blinding were not applicable because of predefined housing conditions. In vivo studies were not blinded and mice were randomly assigned to treatment groups.

## Reporting summary

Further information on research design is available in the Nature Portfolio Reporting Summary linked to this article.

## Data availability

The cryo-EM maps and atomic models were deposited to the EM Data Bank and Protein Data Bank with accession numbers EMD-70721 and PDB 9OPQ (NL63-S$_2$'/HKU1-RBD + TMPRSS2-S441A), EMD-73786 and PDB 9Z3J (NL63-S$_2$' + TMPRSS2-S441A + H1H7 Fab + anti-kappa nanobody), EMD-70722 and PDB 9OPR (H1H7 Fab + anti-kappa nanobody + TMPRSS2-S441A), EMD-75233 (E-FIC + VN01H1 Fab, global refinement), EMD-73656 and PDB 9YYU (E-FIC + VN01H1 Fab, Fab local refinement), EMD-73657 and PDB 9YYV (E-FIC + VN01H1 Fab, S$_2$ local refinement), EMD-73787 and PDB 9Z3K (E-FIC$_{S-v1}$), EMD-75721 (E-FIC$_{S-v3}$ + VN01H1 Fab, global refinement), EMD-75694 and PDB 11HK (E-FIC$_{S-v3}$ + VN01H1 Fab, Fab local refinement), EMD-75695 and PDB 11HL (E-FIC$_{S-v3}$ + VN01H1 Fab, S$_2$ local refinement), EMD-75722 (E-FIC$_{S-v3}$ + C77G12 Fab, global refinement), 75705 and PDB 11HW (E-FIC$_{S-v3}$ + C77G12 Fab, Fab local refinement) and EMD-75697 and PDB 11HN (E-FIC$_{S-v3}$ + C77G12 Fab, S$_2$ local refinement). The MS raw data and search results were deposited to the ProteomeXchange Consortium through the PRIDE partner repository with the dataset identifier PXD069687. Data and materials can be obtained from the corresponding authors upon request and may require a materials transfer agreement. Source data are provided with this paper.

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

## Acknowledgements

This study was supported by the National Institute of Allergy and Infectious Diseases (R01 AI157155 to M.S.D. and P01AI167966, DP1AI158186 and 75N93022C00036 to D.V.), the National Institutes of General Medical Sciences (R00GM147304 to N.M.R,), a Washington Research Foundation Postdoctoral Fellowship (E.S.), FastGrants, an Investigators in the Pathogenesis of Infectious Disease Awards from the Burroughs Wellcome Fund (D.V.) and the University of Washington Arnold and Mabel Beckman cryo-EM center. D.V. is an investigator of the Howard Hughes Medical Institute and the Hans Neurath Endowed Chair in Biochemistry at the University of Washington. We acknowledge the contribution of the scientists and pathologists of NSW Health Pathology and the Kirby Institute to isolate viruses. The funders had no role in study design, data collection and analysis, decision to publish or preparation of the manuscript.

## Author contributions

M.M., J.T.B., L.P. and D.V. conceptualized the project. M.M., J.B.C., J.T.B., J.L., F.L., L.P., M.S.D. and D.V. designed the experiments. M.M. carried out the computational design of E-FIC$_{S-v1}$, E-FIC$_{S-v2}$ and E-FIC$_{S-v3}$ constructs. M.M. vitrified the samples, collected and processed cryo-EM data for the TMPRSS2 structures, as well as E-FIC$_{S-v1}$, E-FIC$_{S-v3}$–VN01H1 and E-FIC$_{S-v3}$–C77G12 structures. M.M. and D.V. built and refined the corresponding atomic models into cryo-EM maps. Y.-J.P. vitrified the VN01H1-bound SARS-CoV-2 E-FIC structure and collected and processed the cryo-EM data. Y.-J.P. and D.V. built and refined the corresponding atomic models into cryo-EM maps. J.T.B. carried out the S cleavage assays and negative-stain imaging and data processing with help from M.M., Y.-J.P., D.A. and D.V. J.T.B. carried out the western blot analysis of cleavage. M.M. designed the NL63-S$_2$'/HKU1-RBD and E-FIC$_S$ proteins. M.M. and M.A.T. designed the DNA constructs. M.M. performed the enzyme kinetics assays. C.G. and C.S. carried out the BLI binding assays. J.L., F.L., D.B. and E.D. performed the neutralization assays. M.M., J.B.C., J.T.B., F.L., J.L., M.S.D. and D.V. analyzed the data. M.M., J.T.B., C.G., C.S. and B.M. recombinantly produced proteins. D.B. bred the mice used for SARS-CoV-2 challenge. J.S.L., A.L. and F.S. provided unique reagents. E.S. and N.M.R. carried out the MS analysis. J.B.C. and S.S. carried out the SARS-CoV-2 challenge studies. B.C. and S.K. carried out SARS-CoV-1 challenge studies. A.A. and S.T. carried out the authentic SARS-CoV-2 neutralization assays using TASL-19 cells. D.V. wrote the manuscript with input from all the authors. M.S.D. and D.V. supervised the project.

## Competing interests

L.A.P. is named as an inventor on patent applications describing the H1H7 antibody and on patents and patent applications for TMPRSS function and models. L.A.P. is a former employee and shareholder of Vir Biotechnology and Regeneron Pharmaceuticals and is an advisor to the AI-driven structure-enabled antiviral platform (ASAP). F.A.L., E.D. and D.B. are current or former employees of Vir Biotechnology and may hold shares in the company. M.M. and D.V. are named on patent applications filed by the University of Washington describing stabilized E-FIC constructs. M.S.D. is a consultant or advisor for Inbios, Vir Biotechnology, IntegerBio, Akagera Medicines and GlaxoSmithKline. The M.S.D. laboratory has received unrelated funding support in sponsored research agreements from Emergent BioSolutions, Generate Biomedicines and Moderna. The other authors declare no competing interests.

## Additional information

**Extended data** is available for this paper at https://doi.org/10.1038/s41594-026-01801-y.

**Correspondence and requests for materials** should be addressed to Michael S. Diamond or David Veesler.

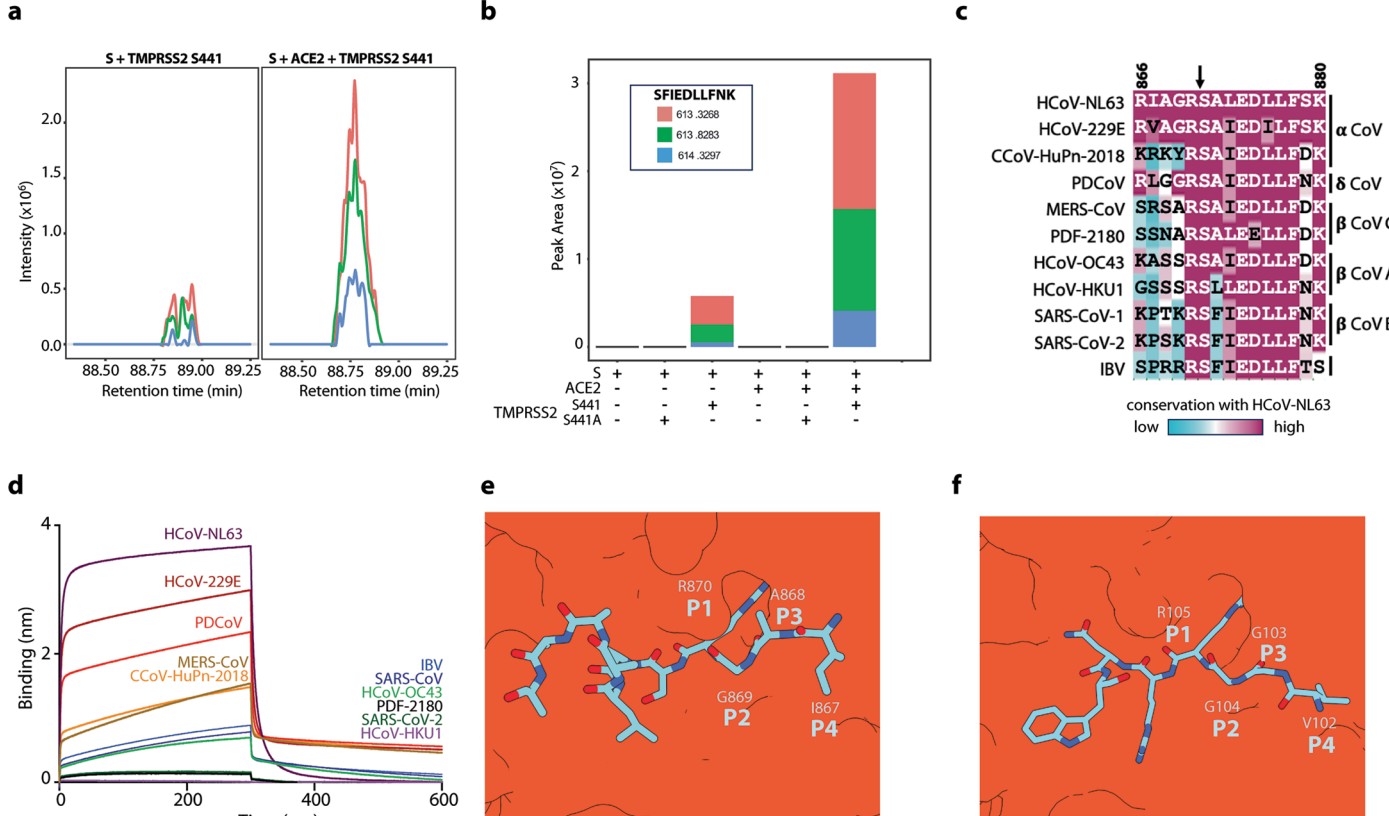

**Extended Data Fig. 1 | Coronavirus S₂′ cleavage, binding and comparison with nanobody A07. a-b**, MS1 chromatograms (**a**) and peak areas (**b**) from the MS/MS analysis of the S$_{816}$FIEDLLFNK$_{825}$ peptide for SARS-CoV-2 S$_{ecto}$ at a concentration of 0.65 μM incubated (or not) with 2 μM of the monomeric human ACE2 (peptidase) ectodomain for 5 minutes prior to incubation (or not) with the TMPRSS2 ectodomain at a concentration of 0.2 μM for 45 min at 4 °C. The samples were digested with GluC as described in the methods section. Red: monoisotopic precursor m/z (M); green (M + 1 m/z), and blue (M + 2 m/z). **c**, Alignment of S₂′ peptide sequences spanning residues 866 to 880 numbered according to the

HCoV-NL63 S sequence. Residues are colored based on their BLOSUM62 similarity to the corresponding residues in HCoV-NL63 S (magenta, high conservation; cyan, low conservation). The S₂′ cleavage site is indicated by a black arrow. **d**, Binding of the human TMPRSS2 S441A (catalytically inactive) ectodomain at a concentration of 5 μM to a panel of biotinylated coronavirus S₂′ peptides immobilized at the surface of BLI SA biosensors. **e**, Close-up view of TMPRSS2 (orange) bound to the HCoV-NL63 S₂′ peptide. **f**, Close-up view of TMPRSS2 (orange) bound to the nanobody A07 (PDB 8SOL).

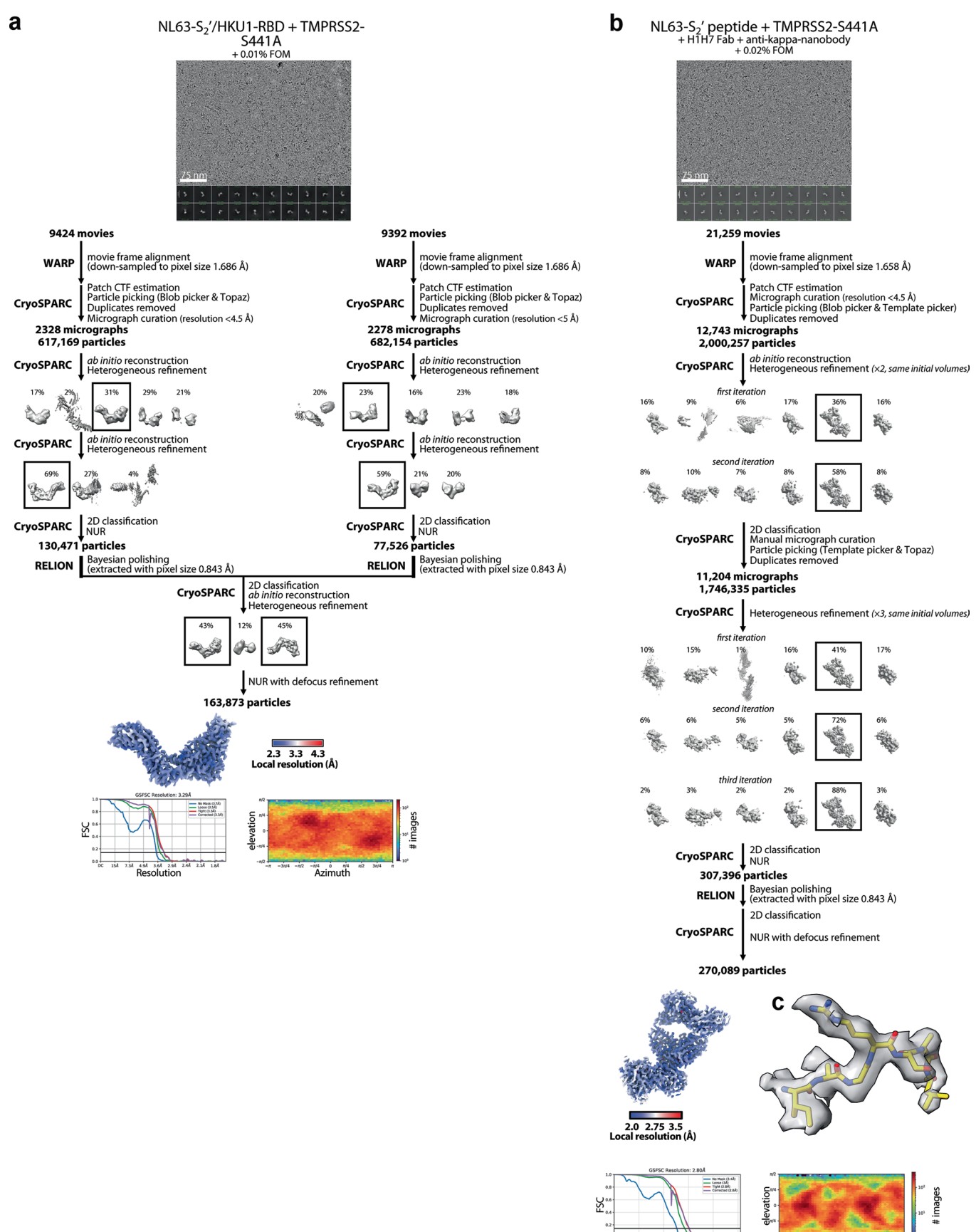

**Extended Data Fig. 2 | See next page for caption.**

**Extended Data Fig. 2 | CryoEM data collection and processing of the NL63-S$_2$'/ HKU1-RBD-bound TMPRSS2 and H1H7-bound TMPRSS2 in complex with the S$_2$' peptide datasets. a-b**, Representative electron micrographs (n = 18,816 for a and n = 21,259 for b), 2D class averages, and data processing workflows for the NL63-S$_2$'/HKU1-RBD-bound TMPRSS2 S441A (**a**) and H1H7(nanobody)-bound TMPRSS2 S441A (**b**) cryoEM datasets. Final reconstructions colored by local resolution calculated in cryoSPARC, gold-standard Fourier shell correlation (FSC) curves with 0.143 cutoff indicated and heat maps of angular distribution of particle orientations are shown at the bottom. CTF, contrast transfer function; NUR, non-uniform refinement. **c**, Zoomed-in view of the S$_2$' peptide with the corresponding region of cryoEM density shown as semi-transparent grey surface from the H1H7(nanobody)-bound TMPRSS2 S441A structure.

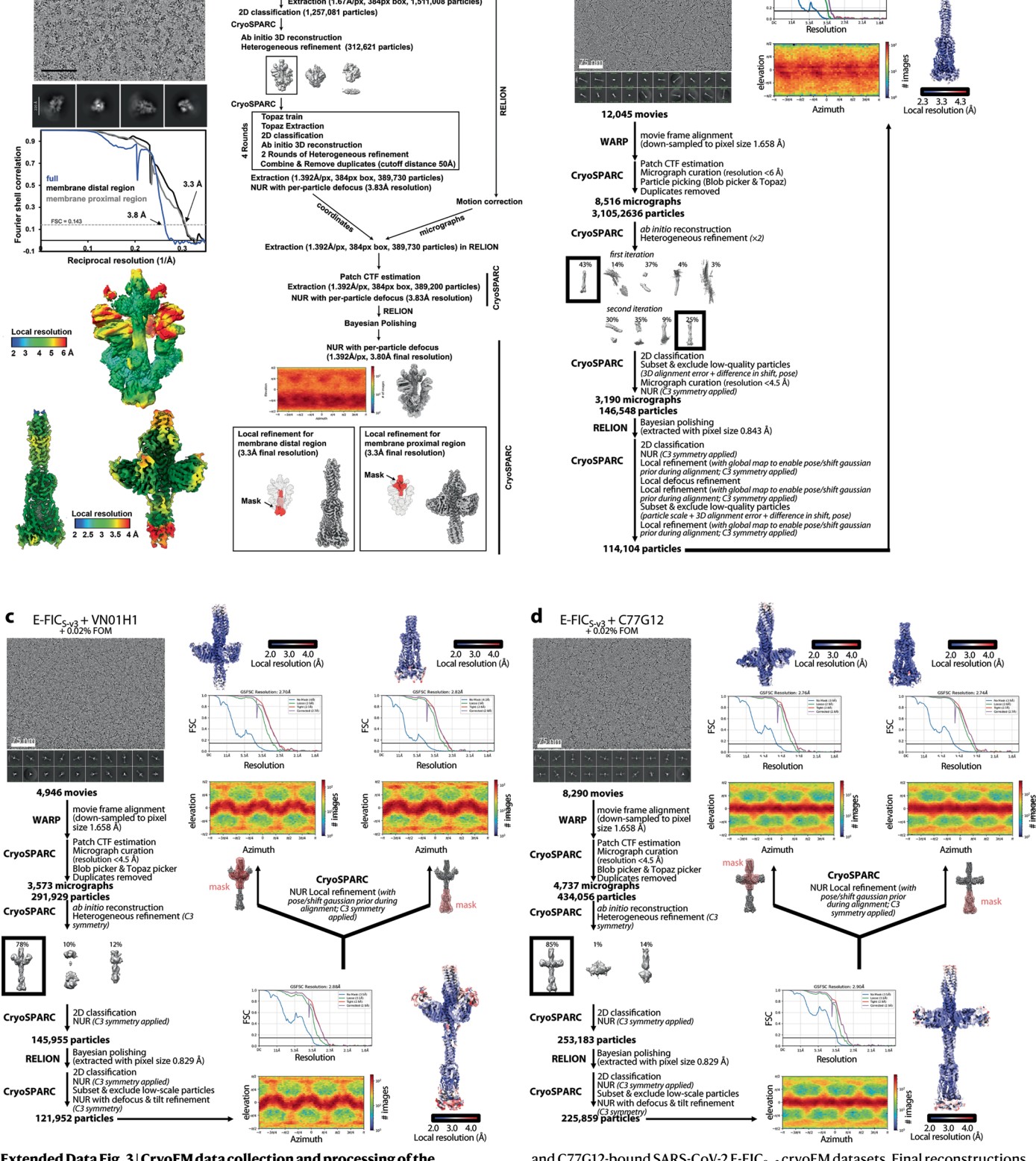

**Extended Data Fig. 3 | CryoEM data collection and processing of the VN01H1-bound SARS-CoV-2 E-FIC, SARS-CoV-2 E-FIC$_{S-v1}$, VN01H1-bound and C77G12-bound SARS-CoV-2 E-FIC$_{S-v3}$ datasets. a-d,** Representative electron micrographs (n = 10,807 for a, n = 12,045 for b, n = 4,946 for c and n = 8,290 for d), 2D class averages, and data processing workflows for the VN01H1-bound SARS-CoV-2 E-FIC, SARS-CoV-2 E-FIC$_{S-v1}$, VN01H1-bound SARS-CoV-2 E-FIC$_{S-v3}$ and C77G12-bound SARS-CoV-2 E-FIC$_{S-v3}$ cryoEM datasets. Final reconstructions colored by local resolution calculated in cryoSPARC, gold-standard Fourier shell correlation (FSC) curves with 0.143 cutoff indicated and heat maps of angular distribution of particle orientations are shown at the bottom. CTF, contrast transfer function; NUR, non-uniform refinement.

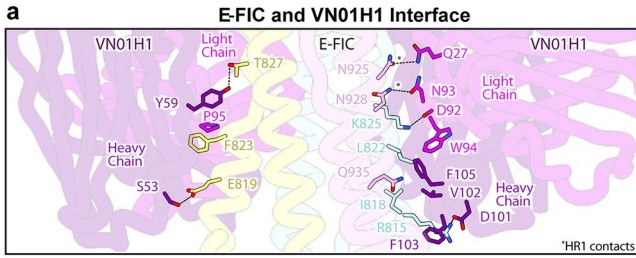

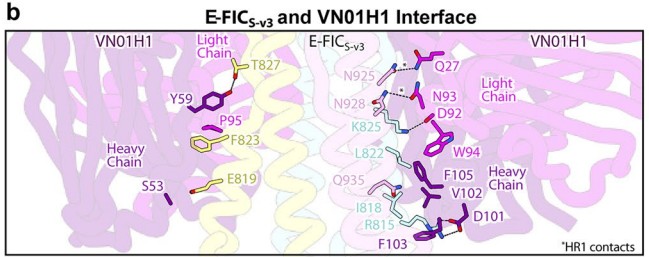

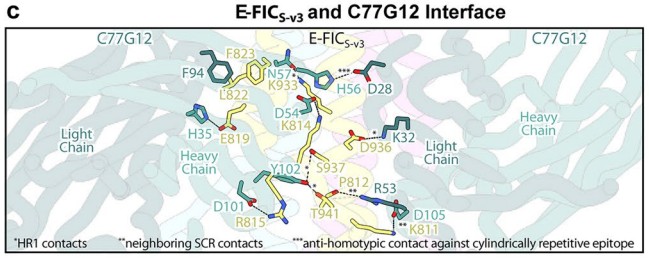

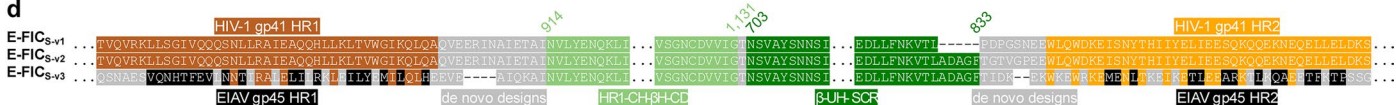

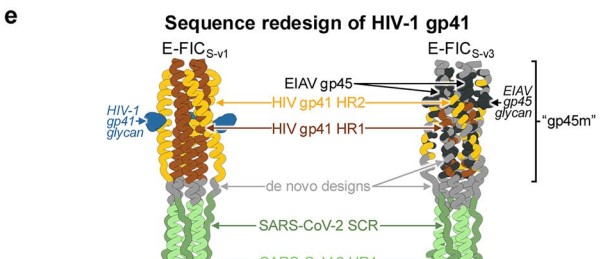

**Extended Data Fig. 4 | Structural comparison of E-FIC and E-FIC$_S$ constructs.** **a-c**, Close-up views of the interfaces between VN01H1 or C77G12 and SARS-CoV-2 E-FIC or E-FIC$_{S-v3}$, with selected interacting side chains shown as sticks and polar interactions indicated by dashed lines. A single asterisk denotes Fab contacts with SARS-CoV-2 S HR1. As E-FIC presents a cylindrically repetitive epitope, a double asterisk denotes Fab contacts with a neighboring SCR from an adjacent chain relative to the primary SCR interaction, while a triple asterisk denotes

anti-homotypic interactions between adjacent Fabs. **d**, Sequence alignment of E-FIC$_{S-v1}$, -v2, and -v3, with residues derived from HIV-1 gp41 HR1 (brown), HIV-1 gp41 HR2 (orange), EIAV gp45 (black), SARS-CoV-2 HR1-CH-βH-CD (light green), and SARS-CoV-2 βUH-SCR (dark green), and de novo–designed regions (grey) indicated. **e**, Structural comparison of the gp41-derived module from E-FIC$_{S-v1}$ (left) and the gp45-derived module from the E-FIC$_{S-v3}$-C77G12 Fab complex (right), colored as in d.

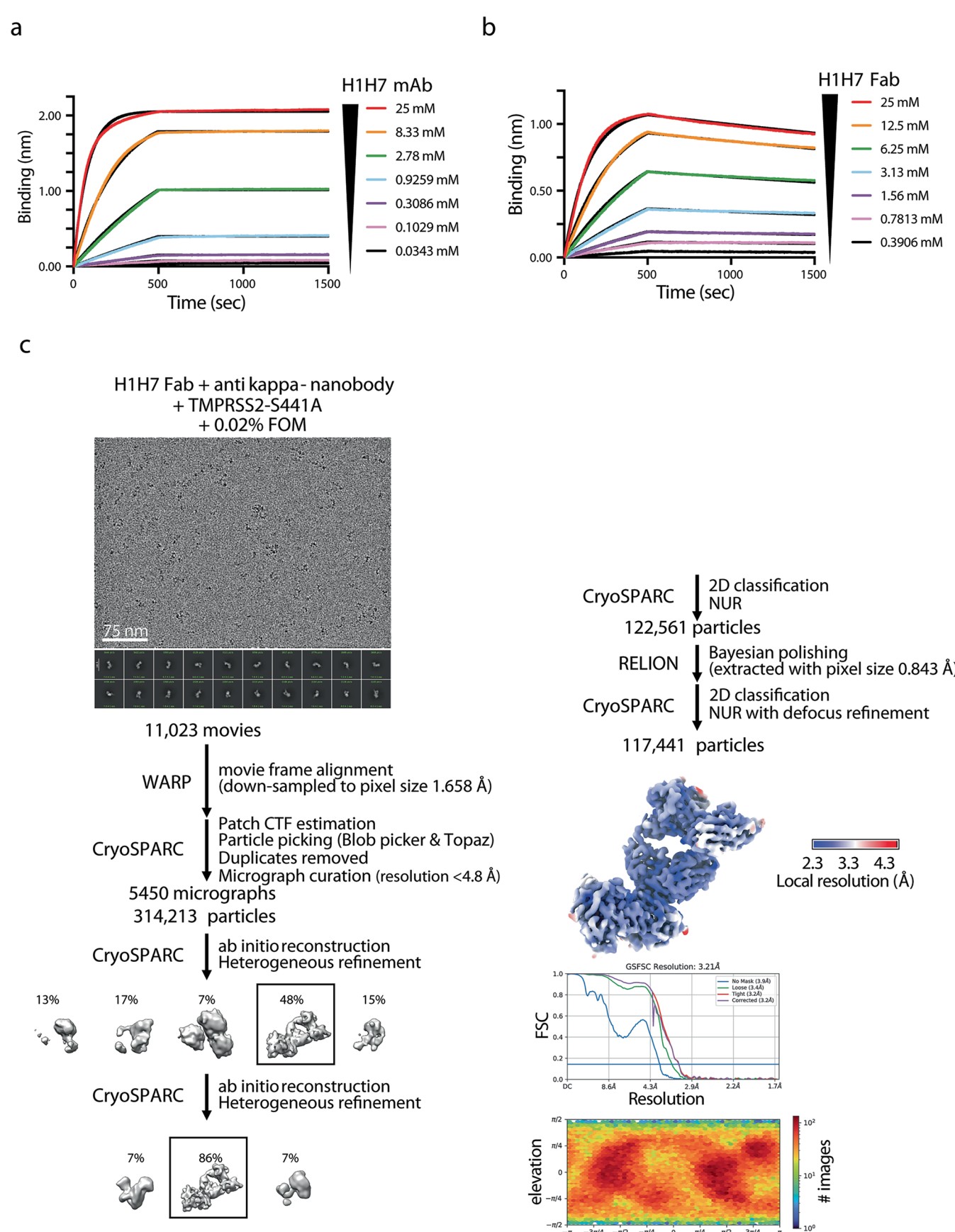

**Extended Data Fig. 5 | See next page for caption.**

**Extended Data Fig. 5 | Biophysical characterization, cryoEM data collection and processing of the H1H7-bound TMPRSS2 dataset. a-b**, BLI analysis of the H1H7 mAb (**a**) or Fab fragment (**b**) binding to the immobilized TMPRSS2 S441A ectodomain immobilized on streptavidin (SA) biosensors. Data correspond to one biological replicate each comprising two technical replicates using the same batch of proteins. Supplementary Table 3 reports the kinetics and affinity parameters obtained for each of the two technical replicates. **c**, Representative electron micrograph (n = 11,023), 2D class averages, and data processing workflow for the H1H7 (and nanobody)-bound TMPRSS2 S441A cryoEM dataset. Final reconstructions colored by local resolution calculated in cryoSPARC, gold-standard Fourier shell correlation (FSC) curves with 0.143 cutoff indicated and heat maps of angular distribution of particle orientations are shown at the bottom. CTF, contrast transfer function; NUR, non-uniform refinement.

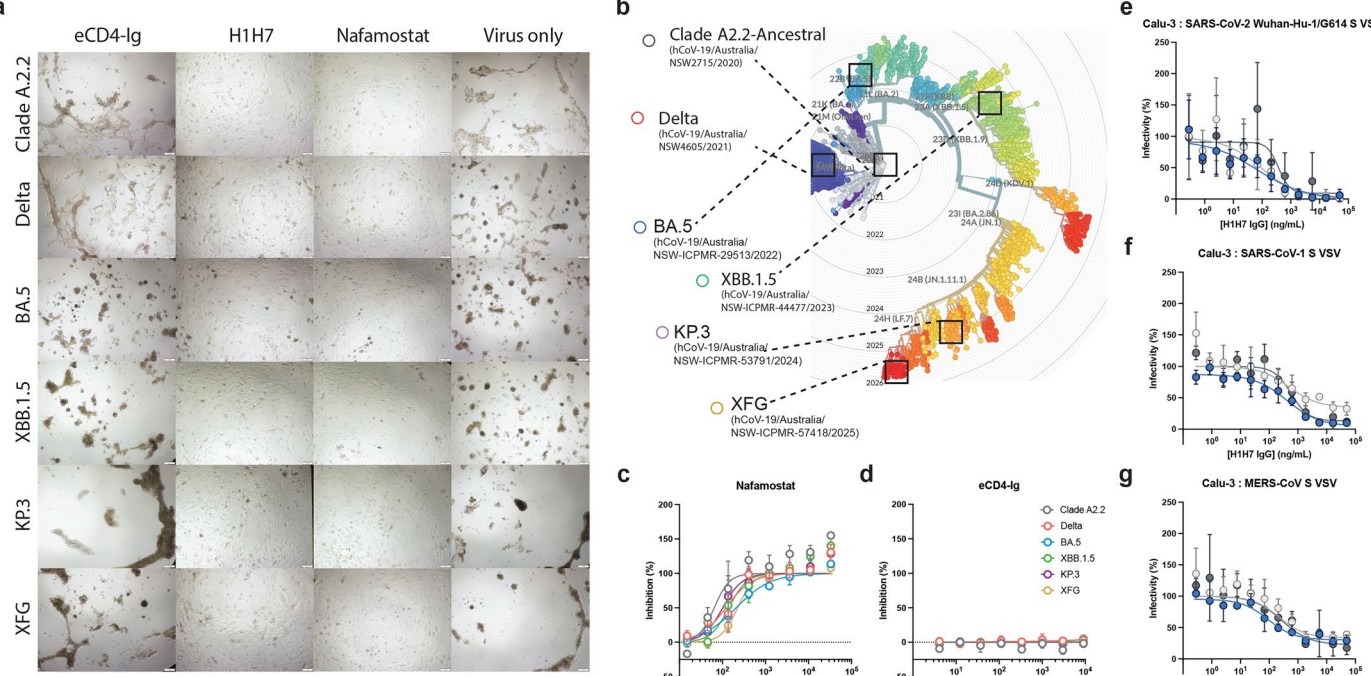

**Extended Data Fig. 6 | Evaluation of H1H7-mediated neutralization of authentic and pseudotyped coronaviruses. a**, Brightfield images of high-throughput SARS-CoV-2 inhibition assay in TASL-19 cells. H1H7 and eCD4-Ig were added at a starting concentration of 9 µg/mL and nafamostat was added at a starting concentration of 11 µM. Scale bar: 50 µm. **b**, Primary SARS-CoV-2 clinical isolates spanning 6 years of the COVID-19 pandemic used in panels e-f are shown on a phylogenetic tree (courtesy of Nexstrain) with their respective GISAID codes indicated. **c-d**, Dose-response curves of Nafamostat- (**c**) and eCD4-Ig- (**d**) mediated inhibition of propagation and syncytia formation of Clade A (Wuhan-Hu-1 related), Delta, BA.5, XBB.1.5, KP.3, and XFG authentic isolates in TASL-19 cells. One representative out of n = 3 (nafamostat) or n = 2 (eCD4-Ig) biological replicates are shown. **e-g**, Dose-response curves of H1H7-mediated inhibition of SARS-CoV-2 Wu/G614 S (**e**), SARS-CoV-1 S (**f**), and MERS-CoV S (**g**) VSV pseudotyped viruses into Calu-3 cells. Each dot is a mean of 3-4 technical replicates and SD shown as lines. Each color corresponds to one biological replicate (n = 3).

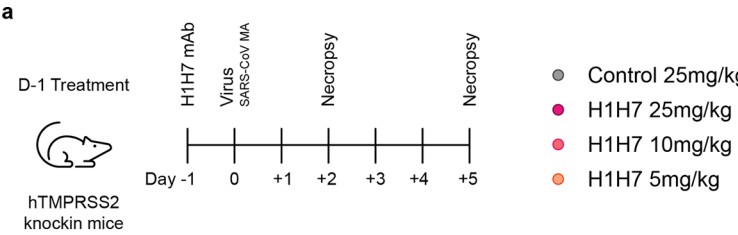

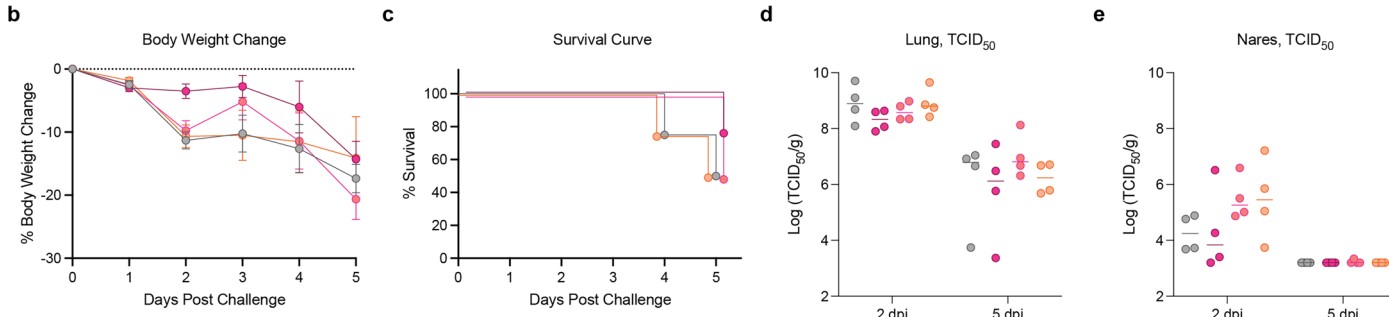

**Extended Data Fig. 7 | Evaluation of the effect of prophylactic administration of the H1H7 mAb on SARS-CoV-1 infection in mice. a**, Study design. 32 human TMPRSS2 KI mice were divided into 4 groups of 8 animals (4 female and 4 male) and were administered 5, 10, or 25 mg/kg of H1H7 mAb or 25 mg/kg isotype control one day prior to intranasal inoculation with $2.5 \times 10^4$ tissue culture infectious dose 50 ($TCID_{50}$) of SARS-CoV-MA15. **b**, Body weight changes throughout the duration of the experiment. Each dot represents the mean values and bars show the SEM. **c**, Survival curve throughout the duration of the experiment. **d-e**, Lung (**g**) and nares (**e**) infectious viral titers were evaluated at 2 and 5 dpi. Median shown as bars.

# Reporting Summary

## Statistics

For all statistical analyses, confirm that the following items are present in the figure legend, table legend, main text, or Methods section.

| n/a | Confirmed | |
|---|---|---|
| ☐ | ☒ | The exact sample size ($n$) for each experimental group/condition, given as a discrete number and unit of measurement |
| ☐ | ☒ | A statement on whether measurements were taken from distinct samples or whether the same sample was measured repeatedly |
| ☐ | ☒ | The statistical test(s) used AND whether they are one- or two-sided<br>*Only common tests should be described solely by name; describe more complex techniques in the Methods section.* |
| ☒ | ☐ | A description of all covariates tested |
| ☐ | ☒ | A description of any assumptions or corrections, such as tests of normality and adjustment for multiple comparisons |
| ☐ | ☒ | A full description of the statistical parameters including central tendency (e.g. means) or other basic estimates (e.g. regression coefficient) AND variation (e.g. standard deviation) or associated estimates of uncertainty (e.g. confidence intervals) |
| ☐ | ☒ | For null hypothesis testing, the test statistic (e.g. $F$, $t$, $r$) with confidence intervals, effect sizes, degrees of freedom and $P$ value noted<br>*Give P values as exact values whenever suitable.* |
| ☒ | ☐ | For Bayesian analysis, information on the choice of priors and Markov chain Monte Carlo settings |
| ☒ | ☐ | For hierarchical and complex designs, identification of the appropriate level for tests and full reporting of outcomes |
| ☒ | ☐ | Estimates of effect sizes (e.g. Cohen's $d$, Pearson's $r$), indicating how they were calculated |

*Our web collection on statistics for biologists contains articles on many of the points above.*

## Software and code

Policy information about availability of computer code

| Data collection | SerialEM 4.2, BioTek Gen5 |
|---|---|
| Data analysis | CryoSPARC 4.6, RELION 3.0 and 5.0, Leginon 3.4, Warp v2.0, Topaz v0.2.5a (implemented in cryoSPARC), ChimeraX 1.10, Coot v0.9.8.93, ISOLDE v1.10.1 (implemented in ChimeraX), Rosetta, Phenix v1.21rc1-5109, GraphPad Prism 10, Sartorius Analysis Software v11.1 |

For manuscripts utilizing custom algorithms or software that are central to the research but not yet described in published literature, software must be made available to editors and reviewers. We strongly encourage code deposition in a community repository (e.g. GitHub). See the Nature Portfolio guidelines for submitting code & software for further information.

## Data

Policy information about availability of data

All manuscripts must include a data availability statement. This statement should provide the following information, where applicable:
- Accession codes, unique identifiers, or web links for publicly available datasets
- A description of any restrictions on data availability
- For clinical datasets or third party data, please ensure that the statement adheres to our policy

The cryoEM maps and atomic models were deposited to the Electron Microscopy Data Bank and Protein Data Bank with accession IDs EMD-70721 and PDB 9OPQ (NL63-S2'/HKU1-RBD + TMPRSS2-S441A), EMD-73786 and PDB 9Z3J (NL63-S2' + TMPRSS2-S441A +H1H7 Fab + anti-kappa-nanobody), EMD-70722 and PDB 9OPR (H1H7 Fab + anti-kappa-nanobody + TMPRSS2-S441A), EMD-75233 (E-FIC + VN01H1 Fab, Global refinement), EMD-73656 and PDB 9YYU (E-FIC + VN01H1 Fab, Fab

# Research involving human participants, their data, or biological material

Policy information about studies with human participants or human data. See also policy information about sex, gender (identity/presentation), and sexual orientation and race, ethnicity and racism.

| | |
|---|---|
| Reporting on sex and gender | N/A |
| Reporting on race, ethnicity, or other socially relevant groupings | N/A |
| Population characteristics | N/A |
| Recruitment | N/A |
| Ethics oversight | N/A |

Note that full information on the approval of the study protocol must also be provided in the manuscript.

# Field-specific reporting

Please select the one below that is the best fit for your research. If you are not sure, read the appropriate sections before making your selection.

☒ Life sciences ☐ Behavioural & social sciences ☐ Ecological, evolutionary & environmental sciences

For a reference copy of the document with all sections, see nature.com/documents/nr-reporting-summary-flat.pdf

# Life sciences study design

All studies must disclose on these points even when the disclosure is negative.

| | |
|---|---|
| Sample size | Sample sizes were based on what is typically used in similar studies, along with our own prior experience. The number of replicates was sufficient to observe consistent and biologically meaningful differences. Key findings were confirmed across independent experiments to ensure reliability. |
| Data exclusions | No data were excluded from the analyses. |
| Replication | All key findings were independently replicated in at least two to three separate experiments using distinct technical and biological replicates. Results were consistent across replicates. |
| Randomization | Randomization was not necessary for this study due to the standardized nature of the experimental setup. |
| Blinding | Blinding was not performed in this study, as it was not relevant to the experimental design. |

# Reporting for specific materials, systems and methods

We require information from authors about some types of materials, experimental systems and methods used in many studies. Here, indicate whether each material, system or method listed is relevant to your study. If you are not sure if a list item applies to your research, read the appropriate section before selecting a response.

## Materials & experimental systems

| n/a | Involved in the study |
|---|---|
| ☐ | ☒ Antibodies |
| ☐ | ☒ Eukaryotic cell lines |
| ☒ | ☐ Palaeontology and archaeology |
| ☐ | ☒ Animals and other organisms |
| ☒ | ☐ Clinical data |
| ☒ | ☐ Dual use research of concern |
| ☒ | ☐ Plants |

## Methods

| n/a | Involved in the study |
|---|---|
| ☒ | ☐ ChIP-seq |
| ☒ | ☐ Flow cytometry |
| ☒ | ☐ MRI-based neuroimaging |

## Antibodies

| | |
|---|---|
| Antibodies used | Anti-SARS-CoV-2 S1 (MAB10540, R&D Systems), anti-SARS-CoV-2 S2 (humanized B6 IgG, described in PMID: 33981021, 10µg/mL and human 76E1, described in PMID: 35773398, 10µg/mL), anti-SARS-CoV-2 S2 polyclonal rabbit antibody (Sino Biological, 1:1000), donkey anti-mouse IgG 680 (LI-COR Biosciences), goat anti-human IgG 680 (LI-COR Biosciences), goat anti-rabbit IgG 680 (LI-COR Biosciences), anti-VSV nucleocapsid antibody (Absolute Antibody), goat anti-mouse IgG Alexa Fluor 647 or 488 (A-21235 or A-11001, Thermo Fisher Scientific), anti-SARS-CoV-2 nucleocapsid antibody (40143-R001, Sino Biological, 1:2000), and goat anti-rabbit IgG Alexa Fluor 647 (A-21245, 1:1000, Thermo Fisher Scientific). |
| Validation | Anti-SARS-CoV-2 S1 (MAB10540, R&D Systems, 2µg/mL) was validated by ELISA and Western blot against the RBD. Anti-SARS-CoV-2 S2 (humanized B6 IgG) was previously characterized (PMID: 33981021) and used here to detect S2 cleavage products. Anti-SARS-CoV-2 S2 polyclonal antibody (Sino Biological) was validated by the manufacturer for Western blot. Anti-VSV nucleocapsid (Absolute Antibody) is widely used in pseudovirus assays to detect VSV-N. Anti-SARS-CoV-2 nucleocapsid (40143-R001, Sino Biological) was validated for immunofluorescence by the manufacturer and used here to detect authentic virus infection. |

## Eukaryotic cell lines

Policy information about cell lines and Sex and Gender in Research

| | |
|---|---|
| Cell line source(s) | Vero E6 cells (CRL-1586, ATCC) were maintained in Dulbecco's Modified Eagle Medium (DMEM) (Invitrogen) supplemented with 10% fetal bovine serum (FBS) (Omega Scientific) and 100 U/mL penicillin-streptomycin (P/S) (Invitrogen). Vero cells expressing TMPRSS2(70 or hACE2-TMPRSS2 (a gift of A. Creanga and B. Graham, National Institutes of Health (NIH)) were maintained as Vero E6 cells, with the addition of 5 µg/mL blasticidin (Vero E6-TMPRSS2) or 10 µg/mL of puromycin (Vero E6-hACE2-TMPRSS2). HEK293T (ATCC, CRL-3216) cells were cultured in DMEM (Gibco) supplemented with 10% FBS (Cytiva) and 1% penicillin-streptomycin (Life Tech). Calu-3 cells (ATCC, HTB-55) were maintained in DMEM/F-12 GlutaMAX™ supplement (Gibco) supplemented with 10% fetal bovine serum (FBS) (Cytiva). All cell lines were maintained at 37°C with 5% CO2. TASL-19 cells were previously described(87). |
| Authentication | Cell lines were not further authenticated. |
| Mycoplasma contamination | Cell lines were not tested for mycoplasma contamination. |
| Commonly misidentified lines (See ICLAC register) | N/A |

## Animals and other research organisms

Policy information about studies involving animals; ARRIVE guidelines recommended for reporting animal research, and Sex and Gender in Research

| | |
|---|---|
| Laboratory animals | This study involved laboratory mice, including hTMPRSS2 knock-in (C57BL/6J-Tmprss2em1(TMPRSS2)Synbl/J) and TMPRSS2 knockout (B6.129-Tmprss2tm1Psn/J) strains, as well as hTMPRSS2-KI mice (C57BL/6Hsd-Tmprss2em1(TMPRSS2)Env). Mice were between 5–8 weeks of age at the time of infection and were housed in HEPA filtered microisolator caging units up to 4 animals per cage at Bioqual. Environmental controls for the animal room were set to maintain 68°F to 79°F, a relative humidity of 30–70%, and a 12hr light/12hr dark cycle. Animals were housed in HEPA filtered microisolator caging units that contained up to 5 animals per cage at Washington University. Environmental conditions for the animal room were set to maintain temperature between 68°F to 74°F, a relative humidity of 30–60%, and a 12h light/12h dark cycle |
| Wild animals | N/A |
| Reporting on sex | Experimental groups included equal numbers of male and female mice to control for sex bias. Sex was considered during study design, and animals were age-matched across groups. However, sex-disaggregated data were not analyzed, as the study was not powered to detect sex-specific effects, and the primary outcomes were not expected to vary by sex. |
| Field-collected samples | N/A |
| Ethics oversight | Ethical approval was obtained for all experiments involving mice. These in vivo studies were conducted under protocols approved by |

| Ethics oversight | the Institutional Animal Care and Use Committee (IACUC) at Washington University and Bioqual. All other experiments involved recombinant protein expression and cell lines and did not require additional ethical oversight. |

Note that full information on the approval of the study protocol must also be provided in the manuscript.

## Plants

| Seed stocks | N/A |

| Novel plant genotypes | N/A |

| Authentication | N/A |

