## [Peer Review file · Nature Structural & Molecular Biology]

TMPRSS2-mediated coronavirus spike activation and inhibition

Corresponding Author: Professor David Veesler

Version 0:

Decision Letter:

Our ref: NSMB-A52560-T

26th Feb 2026

Dear Dr. Veesler,

Thank you for submitting your revised manuscript "TMPRSS2-mediated coronavirus spike activation and inhibition" (NSMB-A52560-T). I am writing to inform you that we'll be happy in principle to publish it in Nature Structural & Molecular Biology, pending minor revisions to satisfy the referees' final requests and to comply with our editorial and formatting guidelines, including further discussion on alternative cell entry mechanisms.

To facilitate our work at this stage, it is important that we have a copy of the main text as a word file. If you could please send along a word version of this file as soon as possible, we would greatly appreciate it; please make sure to copy the NSMB account (cc'ed above).

Sincerely,

Katarzyna Ciazynska, PhD
(she/her)
Senior Editor
Nature Structural & Molecular Biology
<https://orcid.org/0000-0002-9899-2428>

Version 1:

Decision Letter:

31st Mar 2026

Dear Dr. Veesler,

We are now happy to accept your revised paper "TMPRSS2-mediated coronavirus spike activation and inhibition" for publication as an Article in Nature Structural & Molecular Biology.

Your paper will be published online soon after we receive proof corrections and will appear in print in the next available issue. You can find out your date of online publication by contacting the production team shortly after sending your proof corrections.

Authors may need to take specific actions to achieve compliance with funder and institutional open access mandates. If your research is supported by a funder that requires immediate open access (e.g. according to <https://www.springernature.com/gp/open-science/plan-s-compliance> Plan S principles or the <https://www.springernature.com/gp/open-science/us-federal-agency-compliance> NIH public access policy) then you should select the gold OA route, and we will direct you to the compliant route where possible. Because authors warrant under our subscription licensing terms that they haven't committed to licensing any version of their article under a licence inconsistent with the terms of our agreement – including the applicable embargo period – publication under the subscription model isn't suitable for authors whose funders require no embargo.

Sincerely,

Katarzyna Ciazynska, PhD
(she/her)
Senior Editor
Nature Structural & Molecular Biology
<https://orcid.org/0000-0002-9899-2428>

Referee #1 (Remarks to the Author):

While we applaud the effort by the authors to address the reviewers' concerns and revise the manuscript, the central mechanistic question (perhaps the most interesting part of the whole paper) — why and how the S2' cleavage by TMPRSS2 promotes S-mediated fusion — remains unclear.

There are several reports from independent groups showing that ACE2 alone can induce formation of the postfusion conformation of SARS-CoV-2 spike and support membrane fusion (PMID: 39146425; PMID: 37285872; PMID: 40198676), which should not be ignored. In particular, in PMID: 39146425 published in *Science*, it is clearly shown by their fusion assays that membrane fusion can take place at 37°C without adding trypsin or co-expressing TMPRSS2 (Fig.1B in the paper). They also used S-VLP/ACE2-VLP and observed the postfusion conformation of spike after incubating S-VLP with ACE2-VLP without trypsin by cryo-ET (Fig. 3C and 3D). Thus, ACE2 alone can induce the conformational transition of S from the prefusion to postfusion conformation. The conclusion in this *Science* paper is also consistent with PMID: 37285872, a *Nature* paper published by another group, showing that ACE2 alone can induce S1 shedding and transition from the prefusion to postfusion of S2. Likewise, in PMID: 40198676 published in *PLoS Pathogen* by a third group, an HR2 peptide can capture both the S2 fusion intermediate (S2' site not cleaved) and S2' fusion intermediate (S2' site cleaved). Therefore, the cleavage of S2' site by TMPRSS2 does not seem to be a prerequisite for transitioning to the postfusion conformation of S, as claimed in the current manuscript. We agree that it is generally accepted that the S2' cleavage is important for membrane fusion, but increasingly more data are not supporting the conclusion that this specific cleavage is absolutely required for membrane fusion. The author's new data (Figure 3f in the latest version of the revised manuscript) also show that significant binding of E-FICS to HR2 remained (~50%) with E-FICS pre-incubated with inactive (A441) TMPRSS2/no S2' cleavage, in a way, agreeing with the results in PMID: 40198676.

In the revised manuscript, the authors showed that the R815H mutant “had a markedly reduced postfusion S/E-FIC ratio in the presence of ACE2 and TMPRSS2, relative to R815 S”, and believe that “This underscores the importance of the S2' R815 residue, which is strictly conserved in all coronaviruses (Extended Data Fig 7c), for efficient E-FIC refolding to postfusion S”. In a recent paper (PMID: 41309650), many R815 mutants, including R815H, have been shown to have the wildtype fusion activity in both cell-cell fusion and VLP infection assays. The authors should at least test their R815H mutant in their own functional assays to see whether it is still functional or not.

In the rebuttal -- “Fig 4 shows that ~75% inhibition (not 100% inhibition) is reached in VeroE6-TMPRSS2 cells...”, the authors seemed to have not paid attention to the correct figure (Fig 4b, not 4a) and didn't address the point.

In the rebuttal-- “...where both S and ACE2 were embedded in a membrane (PMID:

34930824)”, please check the PMID number since no cryoET data is found in the PNAS paper cited.

We thank the reviewer for their rigorous assessment and the multiple rounds of constructive comments, which helped further refine our manuscript. We appreciate the opportunity to clarify the implications of our findings regarding the role of TMPRSS2-mediated S₂' cleavage in viral entry.

We agree that refolding to postfusion S can be achieved *in vitro* in the presence of ACE2 alone, but only in some specific conditions (typically using a large, non-physiological excess of ACE2, purified proteins, and cell lines not recapitulating *in vivo* conditions of viral entry). We have cited the relevant references, including those from our own group, to reflect this. We would like to remind the reviewer that refolding to postfusion S has also been observed in the absence of receptor (e.g. PMID: 28807998, PMID: 32694201, PMID: 32805734). However, our manuscript distinguishes between what is biochemically possible in isolation and what is physiologically efficient for viral fitness. Our conclusion is not that ACE2-only induced refolding to postfusion S is impossible, but that TMPRSS2-mediated cleavage is the favored route because it markedly enhances the kinetics of the process and in turn viral fitness.

By promoting the refolding of E-FIC to the postfusion S state, TMPRSS2 ensures the rapid, coordinated action of multiple S trimers required for effective fusion pore formation - a nuance supported by the very papers cited by the reviewer:

-PMID: 40198676: “... TMPRSS2 processing of spike to generate S₂' improves [HR2] peptide binding...”.

-PMID: 39146425: “...SARS-CoV-2 virion fusion may require sequential activation processes such as S₂' cleavage by host TMPRSS2 or endosomal cathepsin”.

The revised manuscript comprehensively reviews the literature on S activation and clearly delineates how our work builds on 25 years of coronavirus research that established (i) the importance of S₂' cleavage for viral entry, fusion and fitness (refs 13,14,17,26, 32–35 of our manuscript); and the (ii) key role of TMPRSS2-mediated plasma membrane fusion and entry *in vitro*, *ex vivo* and *in vivo* (refs 33,35–50 of our manuscript). However, the molecular basis of TMPRSS2 recognition, the specific S conformation with which host proteases interact and the conformational consequences of TMPRSS2-mediated S cleavage remained unknown. This knowledge gap has limited our understanding of membrane fusion and hindered efforts to target this critical entry step.

Therefore, our data do not contradict the findings cited by the reviewer but rather integrate them and provide a unified model of coronavirus entry. We delineate a mechanistic framework revealing how TMPRSS2 cleavage lowers the activation barrier for E-FIC-to-postfusion S transition, making it sufficiently efficient to support robust viral entry and pathogenesis *in vivo*.

We would like to re-emphasize the unambiguous and novel conclusions established by the data presented in our manuscript . We reveal that:

1. **TMPrSS2 cleaves and promotes refolding of E-FIC to postfusion S** and provides a **molecular blueprint of these interactions**.
2. **Broadly neutralizing and protective S₂'-directed neutralizing antibodies recognize E-FIC**, which led us to **design a candidate vaccine** against this conformational state.
3. A **TMPrSS2-directed antibody inhibits multiple coronaviruses *in vitro* and protects against SARS-CoV-2** infection in mice.

If TMPRSS2 and S₂' cleavage were not key for viral entry, it would be difficult to explain why S₂'-directed antibodies and the H1H7 TMPRSS2 antibody are neutralizing and protective or why S₂' arginine mutations are deleterious for viral fitness.

Our discovery that the fusogenic conformational changes leading to membrane fusion involve **TMPrSS2-mediated cleavage of E-FIC resolves two conundrums that have challenged the field** since we described the first coronavirus spike structure (Nature, 2016):

1. The coronavirus S₂' cleavage site (e.g. SARS-CoV-2 R815) is inaccessible in prefusion S but becomes exposed in E-FIC.
2. The much closer proximity of S₂' to the host membrane in E-FIC, relative to prefusion S, positions it within reach of TMPRSS2, which is anchored in the host membrane.

Finally, to conclusively address the accessibility of the S₂' site, we have included new data using an improved immunogen, **E-FIC_{S-v3}**.

- We present **two new cryo-EM structures of E-FIC_{S-v3} in complex with distinct S₂'-directed neutralizing Fabs**, recapitulating the native E-FIC structure and confirming the accuracy of our design.
- We added functional assays with E-FIC_{S-v3} showing that **S₂' cleavage is critical for efficient HR2 binding** (Fig. 3), concurring with a large body of literature establishing its importance for viral entry, fusion and fitness.

All changes made to the manuscript are highlighted in blue font along with the references supporting each claim from prior work.

Minor points.

Fig4a-b describe pseudovirus entry assays and authentic virus replication and spread, respectively. TMPRSS2 is key in the propagation of Omicron isolates, a parameter not part of the pseudovirus assays, explaining that the results are not numerically identical between the two assays although both show the same trend and key role of TMPRSS2.

We apologize if a reference was incorrectly inserted in our previous responses. PMID: 40461447 showed that S₂' cleavage is required for membrane fusion to take place using cryoET imaging of authentic SARS-CoV-2 virions mixed with ACE2-harboring pseudoviruses. It was and remains correctly cited in the manuscript.

Referee #2 (Remarks to the Author):

In the revised manuscript, most of my concerns have been addressed, and the authors provided the structure of TMPRSS2 in complex with NL63 S2' peptide, which further supported their main conclusions. However, I still have several concerned issues as following:

(1) Given that the authors have determined the structure of TMPRSS2 in complex with NL63 S2' peptide, the artificial structure of NL63-S2'/HKU1-RBD-TMPRSS2 complex was unnecessary. It would be better to delete related results.

We respectfully disagree as we demonstrated that the HKU1 RBD fusion is a valuable engineering strategy, faithfully recapitulating the structure of the free peptide bound to TMPRSS2, and will be helpful to study peptides with lower affinity than that of HCoV-NL63 (e.g. SARS-CoV-2). It will therefore be useful for the field.

(2) The authors should clarify the molecular mechanism of how TMPRSS2 captures and cleaves the NL63 S2' peptide based on their determined structure. Is there any difference compared to the TMPRSS2 cleavage of host proteins? Is there any common feature for TMPRSS2 cleaving the S2' sites from different coronavirus spike proteins. I think this will be the main novelty of this paper.

The main novelties of the paper are that we elucidate the mechanism of entry of one of the most important groups of pathogens of the century and delineate a novel (possibly pathogen-agnostic) approach to target a conserved host factor to protect against multiple viral families.

TMPRSS2 recognizes its target by cleaving at the scissile bond following the canonical serine protease mechanism. We note that it is the first and only structure of TMPRSS2 in complex with a substrate.

(3) Lines 147-152, in the SI-Fig 9-10, the incubation of R815H S with ACE2 and TMPRSS2 could also induce the formation of post-fusion S at a ratio of 9% or 17%. Why did the authors claim that they failed to detect S₂' cleavage of R815H S? If the R815H S could not be cleaved by TMPRSS2, how could the post-fusion S be formed?

No cleavage could be detected by SDS-PAGE as shown in our paper. As explained in our manuscript and the previous responses to the reviewers' comments, a small fraction of particles may refold to postfusion in the absence of cleavage *in vitro*. However, this process is inefficient and requires S₂' cleavage to occur effectively and enable membrane fusion, concurring with the data shown in Fig1, Fig3g and with prior cryoET work (PMID: 40461447). As stated in the manuscript, this "allows the fusogenic conformational changes of multiple S trimers on a virion to take place in a coordinated manner with TMPRSS2 cleavage promoting efficient refolding to postfusion S and in turn viral entry."

(4) Lines 153-155, if the TMPRSS2 could not cleave the R815H, are there other cleavage sites in the spike protein?

As explained in the discussion section, cathepsin L was proposed to cleave at non-canonical sites although their physiological relevance is unknown (PMID: 35668062).

(5) Fig 3g, in the model, the names of uncleaved and cleaved E-FIC are identical, which might confuse the readers. It would be better to modify the names to distinguish them.

Thank you for this suggestion. We labeled uncleaved and cleaved for clarity in the new panel 3h.

(6) Line 130, Fig SI 7-8 was missing.

Fixed

(7) The authors should show the density of NL63 S₂' peptide.

We added a panel to Extended Data Fig. 8 showing the cryoEM density around the S₂' peptide, as requested.

(8) Line 124, citation error, it would be Fig 1c.

Fixed